# CODA: Learning to Guide Constraint-Aware Optimization for Hardware Accelerators

## Abstract

Designing specialized hardware accelerators is crucial for sustaining the rapid progress of deep learning, yet it remains a costly offline data-driven optimization problem. Evaluating accelerator performance typically requires expensive simulations, while a vast portion of the design space is infeasible due to strict area constraints. Existing data-driven optimization methods often lack feasibility guarantees and suffer from premature convergence on limited offline data. In this paper, we propose the Constrained Offline Design of Accelerators (CODA), a framework for constraint-aware optimization from offline data. CODA tackles data sparsity through a unified surrogate equipped with a cascaded prediction architecture. The model is jointly trained to first predict feasibility and subsequently performance, encouraging the learning of a shared representation that ensures reliable constraint-performance prediction from the limited data. Further, a constraint-aware evolutionary search guided by the surrogate balances exploration and exploitation, accelerating convergence toward feasible high-performance solutions. Extensive experiments on real-world accelerator tasks demonstrate that CODA consistently outperforms both online optimization by 1.11× and surpasses state-of-the-art offline methods by 1.42×. These results highlight CODA as a scalable and robust approach toward automated accelerator design in offline settings. The code is available at https://anonymous.4open.science/r/CODA-38F4/.

## 1 Introduction

Deep learning has driven remarkable advances across diverse domains, including natural language processing (Gao et al., 2025; Wang et al., 2025), image processing (Luo et al., 2024; Yu et al., 2024), disease diagnosis (Zhou et al., 2024), and healthcare (Poulain & Beheshti, 2024). These breakthroughs have been fueled by the availability of large-scale datasets, increasingly complex model architectures, and most importantly, the rise of specialized hardware accelerators such as GPUs (NVIDIA, 2025). Hardware accelerator design, a specialized subset of chip design, focuses on designing dedicated hardware modules to accelerate specific computational tasks. Such accelerators have become pivotal for deep learning applications, offering architectures tailored to maximize computational efficiency and energy effectiveness.

However, designing hardware accelerators is a highly costly optimization problem, as performance evaluation typically relies on simulators that require large amounts of data and numerous queries (Wu & Xie, 2022). Consequently, traditional optimization approaches suffer from inefficiency due to their reliance on numerous expensive evaluations. Furthermore, a large portion of the design space is infeasible due to constraints such as area, power, or energy budgets (Flynn & Luk, 2011; Hegde et al., 2021), and distinguishing feasible from infeasible designs without actual execution is particularly challenging.

Data-driven optimization has recently emerged as a promising direction for addressing expensive optimization problems (Pastrana et al., 2025; Li et al., 2024; Jin et al., 2018). These approaches typically construct surrogate models from offline data to guide the optimizer toward promising regions of complex design spaces. While effective in reducing evaluation costs, they still face several limitations when applied to hardware accelerator design. First, most existing approaches lack explicit, tailored mechanisms to enforce feasibility, leading to wasted evaluations on invalid architectures (Yazdanbakhsh et al., 2022). Second, hardware accelerator design suffers from the problem

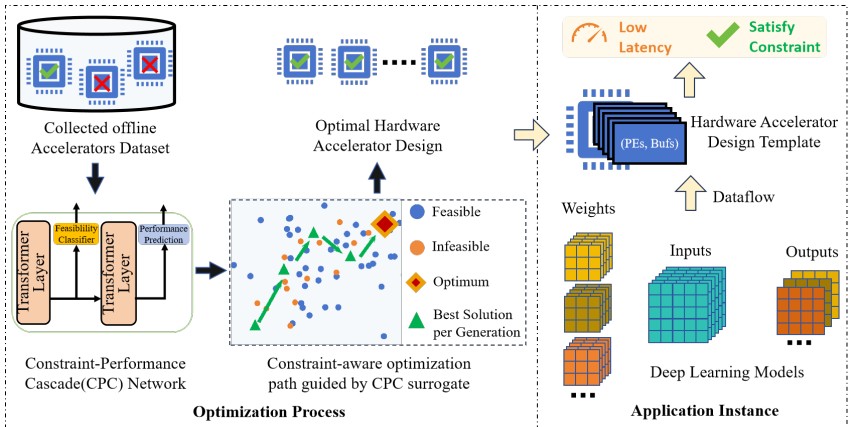

Figure 1: The pipeline of CODA. **Left**: The CPC network is trained on offline datasets and subsequently guides the constraint-aware optimization process. **Right**: A schematic illustration of a simple accelerator architecture design instance, with further details provided in Appendix A.

of data sparsity due to the discrete search space and the use of one-hot embedding, which brings challenges such as low computational efficiency, risk of overfitting, and limited information density (Nardi et al., 2019; Hong et al., 2023). Third, many surrogate-guided methods rely on gradient-based optimization, which is ill-suited for accelerator design due to the discrete and combinatorially complex search space (Trabucco et al., 2021; Qian et al., 2025). Finally, when the target accelerator cases change or a new case is added, the conventional simulation-driven design procedure must be repeated, requiring costly and time-consuming simulations. This substantially increases the overall design time and limits scalability to evolving accelerator architectures (Shi et al., 2020).

In response to the challenges outlined above, we propose the *Constrained Offline Design of Accelerators (CODA)*, a framework for the hardware accelerator design, as illustrated in Fig. 1. Our main contributions are summarized as follows: 1) **Cascaded Surrogate for Constraint-Performance Prediction:** We propose a constraint-performance cascade (CPC) network that integrates a feasibility classifier and a performance predictor in a unified surrogate model. This cascaded architecture provides a robust mechanism to avoid wasted evaluations on invalid designs, ensuring a more efficient exploration of the design space. 2) **Uncertainty-Weighted Dual-Task Learning:** We implement a collaborative dual-task learning approach that jointly trains two encoders to predict feasibility and performance on the same limited dataset. By incorporating uncertainty weighting, this strategy effectively captures complementary aspects of the design space, enriching the shared representations while mitigating the impact of data sparsity. 3) **Constraint-Aware Evolutionary Search:** We develop a constraint-aware evolutionary search where the feasibility classifier serves as a hard filter, and the performance predictor acts as a soft optimization objective. We also design mechanisms that retain some infeasible solutions during the search process, filtering them out in the final output. This approach facilitates fast convergence while maintaining solution diversity. 4) **Zero-Shot Accelerator Design:** We represent accelerator designs as network topologies and develop an offline optimization framework that generalizes effectively to unseen cases, as previously introduced in Yazdanbakhsh et al. (2022). Our approach demonstrates the robustness of this framework in strong zero-shot performance and high data reuse.

Experiment results show the CODA is effective across diverse accelerator cases. Compared to the best designs in the offline training data, CODA achieves an average improvement of **1.11×**. It further outperforms state-of-the-art offline optimization methods by **1.42×** to **8.35×**, and generalizes to unseen cases without test data access, yielding up to **1.65×** improvement over online methods. Overall, CODA enhances both feasibility and performance efficiency of offline hardware accelerator design.

## 2 RELATED WORK

Traditional studies in this field mainly relied on electronic design automation algorithms and methodologies covering various aspects of hardware accelerator design (Khailany, 2020). To tackle

severely increased automation work, more hardware accelerator design work was offloaded from manual effort to machine learning-based automation. For example, MAGNet (Venkatesan et al., 2019) uses Bayesian optimization to explore the complex multidimensional design space. Khailany (2020) also augmented by integrating supervised learning models such as XGBoost, to predict performance and power directly from workload characteristics, drastically reducing the need for iterative simulations and synthesis runs. DOSA (Hong et al., 2023) transforms the design space exploration problem into a differentiable optimization task, enabling fast continuous optimization via gradient descent. HyperMapper (Nardi et al., 2019) targets compiler optimization by continuously interacting with the simulator in a design space with relatively few infeasible points. In contrast to these online methods that rely on expensive simulations during optimization, CODA operates purely offline. It eliminates the need for active simulator queries during the search, enabling scalable exploration with tiny simulation cost at the inference process.

Although machine learning has proven effective in building fast predictive models, reliance on costly simulation and queries remains unavoidable, posing a major challenge for accelerator design. To mitigate this issue, offline data-driven approaches are proposed that reduce dependency on expensive simulations. COMs (Trabucco et al., 2021) proposes a conservative model that lower bounds the actual value of the ground-truth objective on out-of-distribution inputs. DDEA-PF and DDEA-SPF (Huang & Wang, 2021) integrate constraint-handling strategies with data-driven evolutionary algorithms to address constrained optimization tasks. CARCOO (Lu et al., 2023) incorporates the degree of constraint violation into the risk assessment and considers solutions that violate the constraints as high risk. COOREM (Zhu et al., 2025) simultaneously addresses out-of-distribution issues and constrained issues to find stable, high-quality solutions. These general offline algorithms employ trivial, domain-agnostic methods for constraint handling. In contrast, by utilizing a constraint-performance cascade (CPC) model, our CODA captures the topological dependencies of accelerator layers, which is a critical structural nuance that general-purpose BBO methods fail to model.

PRIME (Yazdanbakhsh et al., 2022) is the closest existing work, representing the first attempt to apply offline learning to hardware accelerator design. However, whereas PRIME relies on conservative objective estimation to guide its search, our CODA framework introduces a novel constraint-performance cascade architecture with dual-task learning. This approach enables a more fine-grained disentanglement of feasibility and latency representations during feature learning. As a result, CODA explicitly distinguishes between high-performance designs and those that are truly manufacturable, leading to more reliable and practical optimization outcomes.

## 3 HARDWARE ACCELERATOR OPTIMIZATION PROBLEM

We study the problem of offline hardware accelerator design under strict hardware constraints. Our objective is to identify accelerator architectures that minimize latency while ensuring design feasibility, using only a limited offline dataset. Following prior work (Kao et al., 2020), we parameterize an accelerator architecture design as:

$$x = [(\text{PEs}_1, \text{Buffers}_1), \cdots, (\text{PEs}_n, \text{Buffers}_n)] \tag{1}$$

where $n$ depends on the target deep learning models, $\text{PEs}_i$ specifies the number of processing elements in the $i$-th layer, and $\text{Buffers}_i$ denotes the local buffer capacity, both discretized into 12 levels. A detailed description of the target deep learning models and accelerator architecture defined in this paper is provided in Appendix A.

Let $\mathcal{X}$ denote the design space, where each $x \in \mathcal{X}$ represents a candidate accelerator architecture. In practice, $\mathcal{X}$ is high-dimensional, discrete, and combinatorially complex, with its dimensionality scaling with the number of layers $n$. For a given architecture $x$, we define the performance metric $f(x)$ as its inference latency, which is evaluated using the efficient analytical model MAESTRO (Kwon & et al., 2019). The hardware accelerator optimization problem can then be formalized as:

$$\min_{x \in \mathcal{X}} f(x) \quad \text{s.t. } feasible(x) = 1$$

$$feasible(x) = \begin{cases} 1, & \text{if } x \text{ satisfies area constraints} \\ 0, & \text{otherwise} \end{cases} \tag{2}$$

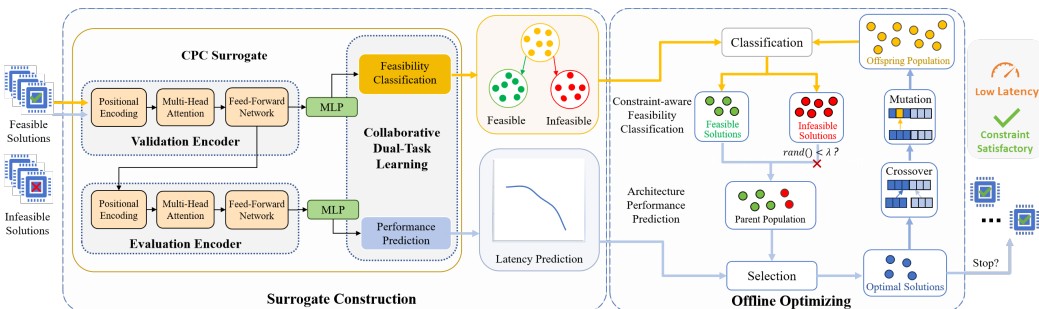

Figure 2: Overview of our CODA. **Left**: Surrogate construction phase, where the Constrained-Performance Cascade (CPC) network is constructed, including its components and network architecture. **Right**: Offline optimization phase, where a constraint-aware evolutionary algorithm leverages the CPC network to efficiently search for feasible, high-performance accelerator designs.

where $feasible(x)$ is a binary indicator denoting whether design $x$ satisfies the area constraint. Infeasible solutions may arise from violations of area, memory, or power constraints, or from mapping and compilation failures. In this work, we primarily focus on area constraints, though the proposed framework can be naturally extended to incorporate other feasibility conditions.

Importantly, evaluating both $f(x)$ and $feasible(x)$ requires queries to analytical models, and in practice may involve costly simulation or synthesis. Moreover, feasible solutions are sparsely distributed among a large portion of infeasible ones (Yazdanbakhsh et al., 2022).

## 4 METHODOLOGY

Our CODA framework consists of two main phases, as illustrated in Fig. 2. During surrogate construction, we leverage limited offline data to model the relationships among accelerator components using a Constrained-Performance Cascade (CPC) network (Sec. 4.1). The CPC network serves three purposes: first, it identifies infeasible designs, reducing redundant evaluations and enabling rapid localization of feasible architectures within the vast design space; second, it provides high-precision latency predictions, allowing the optimizer to concentrate on the most promising candidates; third, the collaborative dual-task learning mechanism alleviates data sparsity by extracting complementary information from feasibility and performance tasks, thereby enhancing the learned representations and improving data efficiency. In the offline optimization, a constraint-aware evolutionary algorithm explores the design space under the guidance of the CPC network (Sec. 4.2). Infeasible individuals with low prediction latency are filtered out, while feasible, high-performing architectures are selected for further evaluation, ensuring both efficiency and effectiveness in the search process.

### 4.1 CONSTRAINED-PERFORMANCE CASCADE NETWORK

**Feasibility Classification.** The input to the CPC network is a candidate accelerator design $x$, represented as a sequence of length $n$ that corresponds to the $n$ layers of the target DNN. Each element is a discrete tuple $(\text{PEs}_i, \text{Buffers}_i)$, which is first encoded as a one-hot vector and then projected into a continuous, learnable embedding space via a trainable embedding layer.

The embedded sequence is then processed by the validation encoder, a self-attention block (Vaswani et al., 2017), to model structural dependencies and interactions among hardware components. In this validation encoder, we employ positional encoding to inject topological prior, as data flows sequentially from layer $i$ to $i+1$ and resource allocation in one layer impacts subsequent stages. Then, the multi-head attention mechanism captures global dependencies, allowing each layer's representation to dynamically attend to others and model complex inter-layer effects such as data reuse and pipeline stalls. This produces a feature-rich embedding, $h_{\text{val}}$, for the downstream task:

$$h_{\text{val}} = \text{ValidationEncoder}(x) \tag{3}$$

Since evaluating infeasible solutions can incur costly redundant simulations, it is important to identify infeasible architectures early in the optimization process. To this end, we introduce a constraint

classification head on top of the validation encoder. Specifically, the output of the validation encoder, $h_{\text{val}}$, is fed into a linear layer, denoted as $\text{MLP}_1$, which transforms the embedding into a scalar value. This scalar value is then passed through the sigmoid function to produce the feasibility probability within $[0, 1]$, interpreted as the likelihood of an architecture being feasible. This formulation enables direct optimization using the binary cross-entropy loss:

$$\hat{y}_f = \sigma(\text{MLP}_1(h_{\text{val}})),$$
$$\mathcal{L}_{\text{feas}} = -y_f \log(\hat{y}_f) - (1 - y_f) \log(1 - \hat{y}_f) \tag{4}$$

where $y_f \in \{0, 1\}$ denotes the ground-truth feasibility label.

The constraint classifier serves as the first gate in the CPC network, efficiently filtering out infeasible architectures and passing only promising candidates to the subsequent performance prediction.

**Performance Prediction.** The architectures predicted as feasible are passed to the performance prediction stage of our CPC network. This stage refines the shared embedding $h_{\text{val}}$ from the validation encoder (Eq. 3) by processing it through an evaluation encoder.

$$h_{\text{eval}} = \text{EvaluationEncoder}(h_{\text{val}}) \tag{5}$$

This cascaded connection serves as a feature reuse mechanism: the validation encoder captures structural feasibility features, while the evaluation encoder refines this representation to isolate features predictive of performance, rather than recomputing from scratch.

Subsequently, a linear layer, $\text{MLP}_2$, regresses this refined embedding to predict the architecture's latency, $\hat{y}_p$. The predictor is trained with a mean squared error loss:

$$\hat{y}_p = \text{MLP}_2(h_{\text{eval}}),$$
$$\mathcal{L}_{\text{perf}} = 1/2 \, (y_p - \hat{y}_p)^2 \tag{6}$$

By reusing the validation encoder's output $h_{\text{val}}$, our surrogate learns a consistent and parameter-efficient representation for both feasibility and performance prediction. This cascaded design effectively steers the optimization toward the identification of high-quality, low-latency designs.

**Uncertainty-Weighted Dual-Task Learning.** Surrogate modeling for offline hardware accelerator design is challenged by the fundamentally different nature of its training signals. Feasibility, a binary outcome, is often sparsely distributed in the vast design space. In contrast, the performance metric is continuous and exhibits wide variations across task configurations.

A static weighting of their respective losses can cause the simpler or less noisy task to dominate the training process, leading to a suboptimal surrogate model. This issue is further compounded by the high-dimensionality of one-hot embeddings, which introduces data sparsity and computational inefficiency.

To address these challenges, we frame the training as a multi-task learning problem with uncertainty-based adaptive loss weighting, inspired by Kendall et al. (2018). The total loss is formulated as:

$$\mathcal{L}_{\text{total}} = \frac{1}{2\sigma_f^2} \mathcal{L}_{\text{feas}} + \frac{1}{2\sigma_p^2} \mathcal{L}_{\text{perf}} + \log \sigma_f + \log \sigma_p \tag{7}$$

where $\mathcal{L}_{\text{feas}}$ and $\mathcal{L}_{\text{perf}}$ are the feasibility and performance losses (Eqs. 4 and 6). The terms $\sigma_f$ and $\sigma_p$ are learnable parameters that represent the homoscedastic task uncertainty of feasibility and performance tasks, respectively. This formulation is theoretically grounded in maximizing the Gaussian likelihood of the dual objectives. The $\log \sigma$ terms act as regularizers, preventing the uncertainties from growing indefinitely and enabling the model to dynamically balance gradient contributions based on each task's inherent noise and scale, without manual tuning.

During training, the model learns to down-weight a task by increasing its corresponding $\sigma$, effectively balancing the two objectives based on their relative confidence. This collaborative dual-task learning with an adaptive weighting mechanism offers several benefits: **1)** By training a shared encoder on two distinct tasks, we impose a strong inductive bias that compels the model to learn fundamental structural features relevant to both feasibility and performance. This multi-view supervision increases the information density extracted from each data point, mitigating overfitting in the sparse data regime. **2)** The adaptive weighting prevents the typically "easier" feasibility task from dominating the gradient updates, thereby ensuring that the more challenging performance regression task is sufficiently learned. This regularization is vital for maintaining generalization capability across the vast design space.

## 4.2 Constraint-Aware Evolutionary Search

In the offline optimization phase, the CPC network replaces costly hardware simulations by serving as an efficient fitness function to guide the search algorithm. However, the hardware accelerator design space presents a significant challenge due to its discrete nature and the complex interplay of architectural parameters. This complexity renders traditional optimization methods ineffective, while the vastness of the design space makes exhaustive search computationally prohibitive. To address these challenges, we propose a constraint-aware evolutionary search strategy that leverages the predictive capabilities of our CPC network to efficiently explore the design space.

Our approach iteratively refines a population of $N$ candidate designs over $T$ generations. Each generation proceeds as follows: **1)** The feasibility classifier filters candidates using a threshold $\theta$, retaining only those architectures that are likely to satisfy constraints. **2)** Since feasible solutions are sparsely distributed and interwoven with infeasible ones, even high-performing infeasible candidates may contain valuable guidance toward optimal feasible designs. To further maintain population diversity and exploit structural information embedded in infeasible solutions, a small fraction of infeasible candidates is stochastically accepted into the next generation with probability $\lambda$. This mechanism balances exploitation of feasible high-quality solutions with exploration of infeasible yet informative regions, thereby improving search efficiency. **3)** For each selected candidate, latency is estimated using the CPC network. **4)** Subsequently, the top-ranked individuals evaluated by the performance predictor undergo crossover and mutation to generate offspring for the next generation. **5)** We maintain an external archive $\mathcal{B}$ with $|\mathcal{B}|$ best designs according to the CPC surrogate, which is updated in each generation. The archive would undergo ground-truth evaluation at the end of the entire algorithm. Note that the archive size is not a parameter of our algorithm, but it serves as a constraint for the final evaluation process, to ensure a fair comparison of different algorithms.

The above process iterates until a termination condition is met. At this point, the best feasible candidates from the final archive $\mathcal{B}$ are validated through ground-truth evaluation to determine the optimal hardware accelerator design. A detailed pseudo-code description is provided in the Appendix B.

## 5 Experimental Results

In this section, we delve into the following research questions: **RQ1**: Can CODA design application-specific accelerators that outperform the best architectures observed in the offline training dataset, while remaining comparable to or surpassing state-of-the-art offline methods under a fixed simulator-query budget? **RQ2**: Can CODA generalize across different accelerator dataflows, leveraging prior offline knowledge to accelerate search in previously unseen applications? **RQ3**: How effective is the CPC network in capturing both constraint and performance trends? **RQ4**: How do the hyperparameters and individual components of CODA influence its overall effectiveness? Below, we first introduce the experimental setups and then discuss RQ1 $\sim$ RQ4 respectively.

**Specific Accelerator Applications.** We conduct experiments on 10 diverse applications, each corresponding to a distinct deep learning model (AlexNet (Krizhevsky et al., 2017), MnasNet (Tan et al., 2018), MobileNetV2 (Sandler et al., 2018), ResNet-50 (He et al., 2016), and ShuffleNet-V2 (Ma et al., 2018)) mapped to a specific dataflow (NVDLA (NVIDIA, 2025) and ShiDianNao (Du et al., 2015)). For clarity, we refer to each accelerator application as Case X, with the corresponding details summarized in Appendix Table 5.

**Baselines.** We compare CODA against four groups of baselines: **1)** *Training-Set Best* ($\mathbb{D}_{Best}$), the best-performing architecture from the training data; **2)** *Online optimization methods*: online GA and online random search (RS). **3)** *Offline optimization methods*: PRIME (Yazdanbakhsh et al., 2022), COOREM (Zhu et al., 2025), CARCOO (Lu et al., 2023), CCOMs (Trabucco et al., 2021), DDEA-PF and DDEA-SPF (Huang & Wang, 2021). The SOO-Bench benchmark platform (Qian et al., 2025) provides evaluation environments for several constrained BBO algorithms, including CARCOO, CCOMs, DDEA-PF, and DDEA-SPF. Further, to evaluate the effectiveness of our CPC network, we compare it with a few **4)** *Classifiers*: Logistic Regression with L1 (LR1) and L2 regularization (LR2) (Ng, 2004), Linear Support Vector Machine (SVM) (Suthaharan, 2016), Random Forest (RF) (Breiman, 2001), and LightGBM (LGBM) (Ke et al., 2017). All hyperparameters of CODA were tuned experimentally, and we list the best settings in the Appendix C.4.

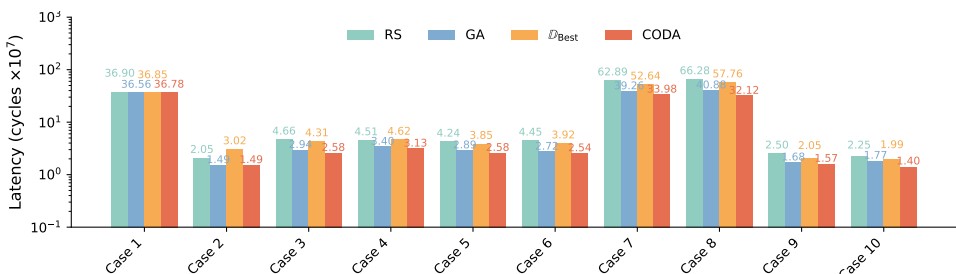

Figure 3: Comparison against training-best designs and online methods (lower is better).

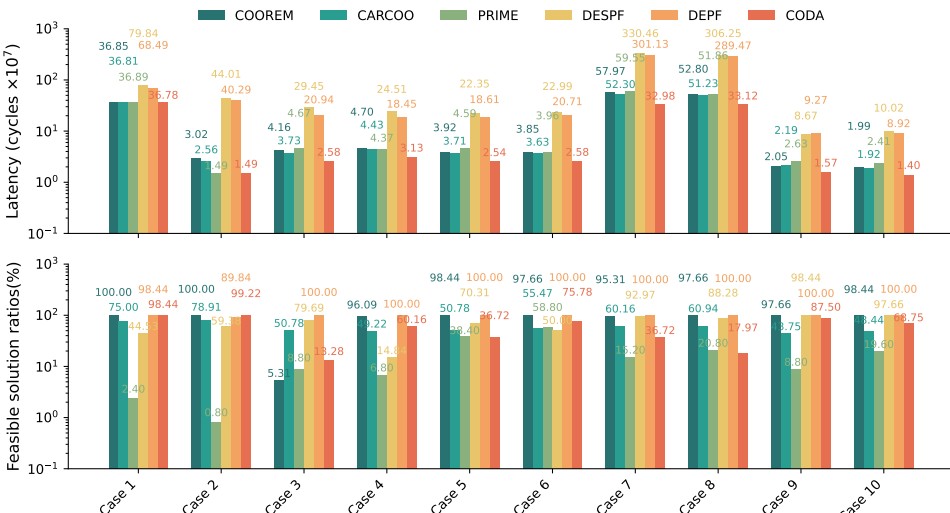

Figure 4: Comparison against offline optimization methods. **Upper**: Optimization results (lower is better); **Lower**: Feasible solution ratios (higher is better).

**Training and Evaluation Protocols.** For each application, 6,000 architectures are randomly sampled for training, while 1,500 architectures are sampled for evaluation. It is worth noting that *online optimization methods*, which are allowed to actively query specific solutions for their objective values, enjoy an inherent advantage over our offline setting. To ensure fairness, we restrict the number of queries to match that of CODA. For the *offline optimization methods*, the $|B| = 128$ final optimal candidates are evaluated with true objective function, along with feasibility. All offline models are trained for 500 epochs with a batch size of 128, and the optimization process is run for 600 generations. More details are provided in the Appendix A.

## 5.1 PERFORMANCE OF CODA IN ACCELERATOR DESIGN (RQ1)

**Comparison with Online Methods.** We compare CODA against the training-best design and representative online optimization methods. As shown in Fig. 3, CODA consistently surpasses all baselines across diverse accelerator design cases. In higher-dimensional model architectures, including MnasNet, MobileNetV2, ResNet50, and ShuffleNetV2 (Cases 3~10), the performance gap becomes particularly pronounced. On average, CODA achieves a geometric mean improvement of **1.59×** over random search and **1.50×** over the best training-set designs. Furthermore, compared with online GA, CODA achieves a stable **1.11×** improvement. It is important to note that CODA is an offline method which is completely unable to use active queries. The results underscore its efficiency.

**Comparison with Offline Methods.** As shown in Fig. 4, CODA achieves pronounced improvements over offline baselines, including **1.50×** over COOREM, **1.42×** over CARCOO, **1.50×** over PRIME, and substantial gains of **8.35×** and **7.29×** over DESPF and DEPF, respectively. Notably, CCOM fails to produce any feasible candidates and is therefore omitted. These results underscore CODA's effectiveness in navigating high-dimensional, constrained design spaces. We also evalu-

ate the proportion of feasible solutions among the final 128 selected candidates (Fig. 4). CODA consistently identifies a large fraction of feasible architectures, with most tasks exceeding **60%** feasibility. While DESPF and DEPF achieve high feasible ratios in many cases, their selected solutions often get trapped in local optima, leading to substantially worse performance. Conversely, PRIME, COOREM, and CARCOO demonstrate limited capability in finding feasible solutions, despite occasionally exploring higher-quality candidates. Overall, these results highlight CODA's ability to simultaneously maintain a high feasible solution ratio and deliver competitive latency performance, emphasizing its practical utility in offline accelerator design.

**Extension to Multi-Constraint Scenarios.** In order to assess CODA's performance under tighter design requirements, we evaluate it in multi-constraint settings involving Area, Power, and Bandwidth. As shown in Appendix C.6, CODA effectively handles the tighter feasible region, demonstrating strong robustness and reliability under more challenging optimization conditions.

## 5.2 ZERO-SHOT OPTIMIZATION ACROSS DATAFLOWS(RQ2)

Many offline dataflows share intrinsic properties across different accelerator design problems, allowing surrogates to capture problem-independent characteristics. In this section, we investigate CODA's zero-shot generalization capability. We introduce a context vector $c_k$ that encodes key properties of each dataflow (See more details of $c_k$ in Appendix C.1). We train CODA on a set of "training dataflows" and then optimize accelerators for unseen "test dataflows" without any queries to the test dataset. Evaluation spans multiple dataflow strategies, including NVDLA, ShiDianNao and Eyeriss (Chen et al., 2016).

Results in Table 1 show that CODA consistently outperforms the offline method PRIME on all problems, and surpasses online optimization methods on higher-dimensional problems such as MnasNet and ResNet50. As the problem scale increases, online methods are limited by iteration budgets and often fail to converge effectively. The CPC network guides the search process more efficiently, enabling deeper exploration and producing reliable, high-quality solutions even in unseen scenarios. For lower-dimensional problems such as AlexNet, online search is competitive, as optimal solutions can be approximated with few evaluations. Nevertheless, CODA achieves all results without accessing test datasets during training, demonstrating that the contextual representations learned from prior applications generalize effectively to unseen dataflows. This highlights CODA's strength in reusing offline accelerator data and adapting to new design scenarios in large, complex search spaces.

Table 1: Zero-shot optimization results across different deep learning models and dataflows. Reported values denote latency in cycles ($\times 10^7$). CODA does not use any additional data from the target applications. Mark "–" in the table indicates that no feasible solutions were found in the final ground-truth evaluations.

| DL Model | Dim. | Train Dataflow | Test Dataflow | Online GA | Online RS | PRIME | CODA |
|---|---|---|---|---|---|---|---|
| AlexNet | 10 | Eyeriss, ShiDianNao | NVDLA | **36.56** | 36.90 | 50.70 | 38.28 |
| AlexNet | 10 | NVDLA, ShiDianNao | Eyeriss | **0.14** | 0.62 | – | **0.14** |
| MnasNet | 104 | Eyeriss, ShiDianNao | NVDLA | 3.40 | 4.66 | 4.63 | **3.22** |
| MnasNet | 104 | NVDLA, ShiDianNao | Eyeriss | 3.26 | 4.85 | – | **3.24** |
| ShuffleNetV2 | 112 | Eyeriss, ShiDianNao | NVDLA | 1.77 | 2.09 | 2.52 | **1.69** |
| ShuffleNetV2 | 112 | NVDLA, ShiDianNao | Eyeriss | 1.50 | 2.50 | 2.60 | **1.42** |
| **Geomean of CODA Improvement** | | | | 1.02 | 1.65 | 1.50* | 1.00 |

*Computed over 4 available cases only, since PRIME failed to find feasible solutions in the other 2 cases.

## 5.3 CPC NETWORK PERFORMANCE (RQ3)

As the core of CODA, the CPC network captures dependencies among accelerator components through a network topology representation. We evaluate our CPC network from two perspectives: feasibility classification and latency prediction.

**Feasibility Classification.** We compute the confusion matrix and report four widely used classification metrics: *Precision* (Prec.), *Recall* (Rec.), *F1-score* (F1), and *Accuracy* (Acc.) on the unseen test dataset. As shown in Table 2 and Fig. 5a, with a fixed threshold of 0.5, the classifier achieves an F1-score of 0.94, accuracy of 0.94, recall of 0.92, and precision of 0.95, demonstrating a strong dis-

Table 2: Comparison results across different applications, with the highest F1-score highlighted in **bold**. Results are reported as mean (standard deviation) across three independent test datasets.

| Case | Application | LR1 F1 | LR2 F1 | SVM F1 | RF F1 | LGBM F1 | CPC F1 | CPC Acc. | CPC Rec. | CPC Prec. |
|---|---|---|---|---|---|---|---|---|---|---|
| Case 1 | AlexNet-NVDLA | 0.86 (0.01) | 0.86 (0.01) | 0.86 (0.00) | 0.96 (0.00) | 0.97 (0.01) | **0.99 (0.00)** | 0.99 (0.00) | 0.97 (0.01) | 1.00 (0.00) |
| Case 2 | AlexNet-ShiDianNao | 0.87 (0.01) | 0.87 (0.01) | 0.87 (0.00) | 0.95 (0.01) | 0.97 (0.01) | **0.99 (0.00)** | 0.99 (0.00) | 0.98 (0.00) | 1.00 (0.00) |
| Case 3 | MnasNet-NVDLA | 0.85 (0.00) | 0.85 (0.00) | 0.85 (0.00) | 0.78 (0.01) | 0.85 (0.01) | **0.93 (0.01)** | 0.93 (0.01) | 0.93 (0.01) | 0.94 (0.01) |
| Case 4 | MnasNet-ShiDianNao | 0.85 (0.01) | 0.85 (0.01) | 0.85 (0.01) | 0.78 (0.02) | 0.85 (0.00) | **0.93 (0.01)** | 0.93 (0.00) | 0.93 (0.01) | 0.93 (0.01) |
| Case 5 | MobileNetV2-NVDLA | 0.86 (0.00) | 0.86 (0.00) | 0.85 (0.00) | 0.79 (0.01) | 0.86 (0.01) | **0.93 (0.01)** | 0.93 (0.01) | 0.88 (0.01) | 0.97 (0.00) |
| Case 6 | MobileNetV2-ShiDianNao | 0.85 (0.01) | 0.85 (0.01) | 0.85 (0.01) | 0.78 (0.01) | 0.85 (0.00) | **0.93 (0.01)** | 0.94 (0.01) | 0.92 (0.01) | 0.95 (0.01) |
| Case 7 | ResNet50-NVDLA | 0.86 (0.00) | 0.86 (0.00) | 0.86 (0.01) | 0.79 (0.02) | 0.85 (0.01) | **0.92 (0.00)** | 0.92 (0.00) | 0.88 (0.01) | 0.96 (0.01) |
| Case 8 | ResNet50-ShiDianNao | 0.86 (0.01) | 0.86 (0.01) | 0.86 (0.01) | 0.77 (0.01) | 0.85 (0.01) | **0.92 (0.00)** | 0.92 (0.01) | 0.90 (0.01) | 0.94 (0.01) |
| Case 9 | ShuffleNetV2-NVDLA | 0.86 (0.01) | 0.86 (0.01) | 0.86 (0.01) | 0.76 (0.00) | 0.84 (0.01) | **0.93 (0.01)** | 0.93 (0.00) | 0.94 (0.01) | 0.91 (0.00) |
| Case 10 | ShuffleNetV2-ShiDianNao | 0.86 (0.01) | 0.86 (0.01) | 0.85 (0.00) | 0.77 (0.01) | 0.86 (0.01) | **0.93 (0.01)** | 0.93 (0.01) | 0.90 (0.01) | 0.96 (0.00) |
| | **Average** | 0.86 | 0.86 | 0.86 | 0.81 | 0.87 | **0.94** | 0.94 | 0.92 | 0.95 |

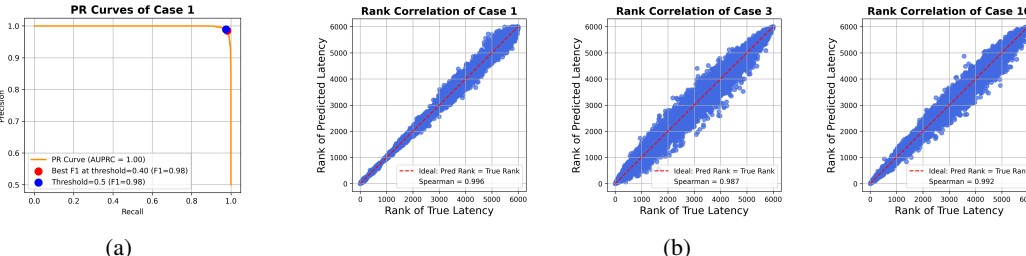

(a)                                    (b)

Figure 5: CPC performance analysis. (a) PR-curve of Case 1; (b) Spearman's rank correlation coefficient of Case 1, 3, 10. See more details in Appendix C.3.

criminative ability between feasible and infeasible architectures across diverse hardware scenarios. Furthermore, CPC consistently attains the highest F1-score compared with other classifiers across varying cases. Detailed results are provided in Appendix Fig. 6.

**Latency Prediction**. We assess latency predictions using Spearman's rank correlation coefficient $\rho$ (Myers et al., 2013) between predicted and ground-truth latency on the unseen test dataset. As shown in the Fig. 5b, CPC achieves high rank correlations, averaging **0.99**, which validates its effectiveness in ranking populations and provides reliable guidance for the search process.

## 5.4 SENSITIVITY ANALYSIS AND ABLATION STUDIES (RQ4)

**Effects of $\theta$ and $\lambda$**. 1) The classification threshold $\theta$ controls the strictness of the feasibility filter. A large value may incorrectly reject feasible candidates, while a small value may allow infeasible ones to pass through. By comparing $\theta = 0.5$ with the thresholds that maximize the F1-score on precision–recall curves, we observe negligible differences ($\leq 0.01$) across all cases. Thus, we adopt $\theta = 0.5$ throughout. 2) For the infeasible-acceptance probability $\lambda$, we vary its value between $0.05$ and $0.5$. In low-dimensional applications, the performance is relatively insensitive to $\lambda$, as most settings lead to comparable optimal solutions. In contrast, in high-dimensional applications, $\lambda$ has a more noticeable influence on solution quality. Nevertheless, $\lambda = 0.1$ consistently achieves the best or near-best performance across the majority of cases, while the performance gap compared to other values remains modest (e.g., the largest gap is only $0.09$ in Case 6). Therefore, we adopt $\lambda = 0.1$ as the default setting in all experiments (see Appendix C.4).

**Sensitivity of Offline Dataset Size**. We justified our choice of 6000 samples through sensitivity analysis and efficiency trade-offs. We conducted an ablation study by training and evaluating CODA with varying offline dataset sizes: 1,000, 3,000, 6,000, and 9,000. As shown in Table 3, 6,000 samples strike the optimal balance. In high-dimensional tasks, small datasets lead to an under-trained CPC model that fails to guide the search to any feasible solution (denoted by "–"). Increasing to 9000 yield only marginal gains (with the best improvement being only 0.18 in Case 7), but increases data collection costs by 50%.

**Ablation Study on CODA**. We evaluate four variants of our method: 1) *CODA-w/o FP*, which removes the feasibility classifier from the CPC network; 2) *CODA-w/o CE*, which simplifies the architecture by eliminating the cascaded encoder; 3) *CODA-w/o UW*, which replaces the adaptive

Table 3: Comparison across different offline training dataset sizes of CODA. Values denote the best latency $(\times 10^7)$, with parentheses indicating the change relative to the 6000-sample baseline. Mark "–" in the table indicates that no feasible solutions were found in the final ground-truth evaluations.

| Case | 1000 | 3000 | 6000 | 9000 |
|---|---|---|---|---|
| Case 1 | 36.56 (+0.22) | 36.64 (+0.14) | 36.78 (0.00) | **36.65 (-0.13)** |
| Case 2 | 1.49 (0.00) | 1.49 (0.00) | **1.49 (0.00)** | **1.49 (0.00)** |
| Case 3 | – | – | **2.58 (0.00)** | 2.62 (+0.04) |
| Case 4 | 3.94 (+0.81) | – | 3.13 (0.00) | **3.08 (-0.05)** |
| Case 5 | – | – | **2.54 (0.00)** | 2.58 (+0.04) |
| Case 6 | – | – | 2.58 (0.00) | **2.53 (-0.05)** |
| Case 7 | – | – | 32.98 (0.00) | **32.80 (-0.18)** |
| Case 8 | 42.56 (+9.44) | 35.74 (+2.62) | **33.12 (0.00)** | 33.89 (+0.77) |
| Case 9 | – | – | **1.57 (0.00)** | 1.57 (0.00) |
| Case 10 | 1.75 (+0.35) | 1.43 (+0.03) | 1.40 (0.00) | **1.38 (-0.02)** |

Table 4: Optimization results (lower is better) across different variants of CODA. Reported values denote latency in cycles $(\times 10^7)$. Mark "–" indicates no feasible solution found.

| Case | CODA-w/o FP | CODA-w/o CE | CODA-w/o UW | CODA-w/o IT | CODA-w/o TR | CODA-w/o ES | CODA |
|---|---|---|---|---|---|---|---|
| Case 1 | – | **36.56** | 36.75 | 36.77 | 36.73 | 36.65 | 36.78 |
| Case 2 | – | **1.49** | **1.49** | **1.49** | **1.49** | **1.49** | **1.49** |
| Case 3 | – | 2.61 | 2.90 | 2.61 | 6.49 | 3.92 | **2.55** |
| Case 4 | – | 3.17 | 3.35 | **3.13** | 5.27 | 4.30 | **3.13** |
| Case 5 | – | 2.63 | 2.91 | **2.58** | 4.32 | 3.70 | **2.58** |
| Case 6 | – | 2.57 | 2.80 | 2.56 | 5.34 | 3.77 | **2.54** |
| Case 7 | – | **32.30** | 39.43 | 33.56 | 80.27 | 54.83 | 32.98 |
| Case 8 | – | 33.79 | 37.95 | 34.86 | 79.06 | 50.96 | **33.12** |
| Case 9 | – | **1.57** | 1.74 | 1.58 | 2.77 | 2.12 | **1.57** |
| Case 10 | – | 1.37 | 1.47 | **1.40** | 2.9 | 1.99 | **1.40** |
| **Geomean of CODA Improvement** | – | 1.0027 | 1.0907 | 1.0095 | 1.7771 | 1.4350 | 1.0000 |

loss weighting with a fixed weight; 4) *CODA-w/o IT*, which disables the infeasible-assistant selection strategy, restricting the evolutionary search to feasible solutions only; 5)*CODA-w/o TR, which substitutes the Transformer-based encoder with a standard MLP encoder of comparable parameter count; and 6)CODA-w/o ES, which replaces the evolutionary search strategy with random search algorithm.* Experimental results in Table 4 show that the complete CODA outperforms all variants in 8 out of 10 cases, indicating that each component is crucial for robust optimization. In particular, *CODA-w/o FP* consistently performs poorly, underscoring the importance of the feasibility classifier in steering the search away from infeasible designs. Meanwhile, adaptive loss weighting and sufficient surrogate capacity further enhance the stability and accuracy of surrogate learning. Notably, the performance of *CODA-w/o TR* and *CODA-w/o ES* degrades substantially on complex, high-dimensional cases. This result confirms that the Transformer encoder is critical for capturing intricate cross-layer dependencies, while the evolutionary search is indispensable for system-level optimization scalability.

## 6 CONCLUSION AND DISCUSSION

This paper proposes CODA for constraint-aware hardware accelerator optimization using offline data. CODA addresses key challenges in this domain, including small feasible regions, data sparsity in large design spaces, and costly simulations. Within a unified modeling framework, CODA constructs a CPC network consisting of a feasibility classifier and a latency predictor. To enhance training under sparse data, we adopt a collaborative dual-task learning mechanism with uncertainty weighting. During the constraint-aware evolutionary search, the feasibility classifier filters out infeasible configurations, enabling the latency predictor to guide the search efficiently toward feasible, high-performing regions without requiring costly ground-truth evaluations. We validate the effectiveness of CODA across a broad range of accelerator tasks. Experimental results demonstrate that CODA and its CPC consistently achieve strong performance. However, we also observed that the CPC's accuracy in predicting constraint satisfaction still has room for improvement (As shown in Appendix D). Future work could be paid to address this issue by exploring refinements in the modeling and/or developing more sophisticated approaches to offline data collection and usage.

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

# A    ACCELERATOR ARCHITECTURES AND DATASETS

**Deep Learning (DL) Models.** DL models are typically constructed from a few fundamental layers. Each layer exhibits distinct computation and memory-access patterns that directly influence accelerator design. For example, convolutional layers dominate in DNNs like ResNet-50 and MobileNetV2, targeting image processing tasks. Fully connected layers or MLPs are often used as the last layer in many DL models. Different layer types offer varying degrees of data reuse opportunities, which can be exploited by accelerators depending on the dataflow. In this paper, we consider widely used models including AlexNet (Krizhevsky et al., 2017), MnasNet (Tan et al., 2018), MobileNetV2 (Sandler et al., 2018), ResNet-50 (He et al., 2016), and ShuffleNet-V2 (Ma et al., 2018). These DL models differ in layer types, number of layers, parameter count, and kernel dimensions, which confirms that our evaluation effectively tests algorithm performance across varied problem characteristics.

**Dataflows.** Dataflow defines how data is orchestrated between global SRAM and local PE buffers, including computation ordering, parallelization across PEs, and tiling strategies. Different accelerators implement distinct dataflow mechanisms to optimize reuse and efficiency. In our experiments, we include representative designs such as NVDLA (NVIDIA, 2025), Eyeriss (Chen et al., 2016), and ShiDianNao (Du et al., 2015), each showcasing unique dataflow strategies.

**Hardware Resources.** Deep learning accelerators design problem consists of an array of Processing Elements(PEs) and Buffers(Bufs) as shown in Fig. 1. Each PE performs partial-sum computations, while each Buffer stores weights, activations, and partial sums. Given a specific choice of PEs, Buffers, and dataflow, we can determine other architectures, such as global shared buffer and Networks-on-Chip. Therefore, we represent the hardware architecture input as a sequence of discrete (PEs, Buffers) in this paper. Following previous work (Kao et al., 2020), Hardware architecture variables are discretized into 12 levels for both processing elements and buffer sizes, as illustrated in Table 5.

Table 5: Parameter value ranges used in the experiments.

| Parameter | Discrete Values |
| --- | --- |
| PEs | 1, 2, 4, 8, 12, 16, 24, 32, 48, 64, 96, 128 |
| Buffers Unit | 1, 2, 3, 4, 5, 6, 7, 8, 9, 10, 11, 12 |

**Dataset details.** We include 10 different accelerator cases in Sec.5 denoted as Case 1∼10, with details summarized in Table 6. According to the problem formulation in Sec. 3, the input is represented as a sequence of discrete (PEs, Buffers) tuples. Since different DL models comprise different numbers of layers, the input dimension is determined by the layer count, while the computational complexity increases exponentially with network depth.

Table 6: Details of the evaluated datasets across different DL models and dataflows, denoted as Case 1-10.

| Case | DL Model | Dataflow | Input Dim. | Area Constraint |
| --- | --- | --- | --- | --- |
| Case 1 | AlexNet | NVDLA | $5 \times 2$ | 22.36 mm$^2$ |
| Case 2 | AlexNet | ShiDianNao | $5 \times 2$ | 22.60 mm$^2$ |
| Case 3 | MnasNet | NVDLA | $52 \times 2$ | 24.23 mm$^2$ |
| Case 4 | MnasNet | ShiDianNao | $52 \times 2$ | 24.19 mm$^2$ |
| Case 5 | MobileNetV2 | NVDLA | $52 \times 2$ | 24.21 mm$^2$ |
| Case 6 | MobileNetV2 | ShiDianNao | $52 \times 2$ | 24.23 mm$^2$ |
| Case 7 | ResNet50 | NVDLA | $53 \times 2$ | 24.69 mm$^2$ |
| Case 8 | ResNet50 | ShiDianNao | $53 \times 2$ | 24.72 mm$^2$ |
| Case 9 | ShuffleNetV2 | NVDLA | $56 \times 2$ | 26.07 mm$^2$ |
| Case 10 | ShuffleNetV2 | ShiDianNao | $56 \times 2$ | 26.10 mm$^2$ |

## B Optimization Details of CODA

For the offline optimization of CODA, the constraint-aware evolutionary search operates in a round-robin manner. The detailed procedure is summarized in Algorithm 1. The search begins by randomly sampling candidate architectures and ranking them according to their predicted latency. Once initialized, the evolutionary iterations proceed with two key steps: 1) **Constraint-Aware Selection**: Feasible solutions are identified by the CPC surrogate according to their estimated feasibility probability. To encourage exploration, a fraction of infeasible solutions is retained for the next generation with probability $\lambda$. Concretely, a random value rand() $< \lambda$, the corresponding infeasible solution survives to the offspring population. 2) **Crossover and Mutation**: To maintain population diversity and explore previously unvisited regions of the search space, new offspring are generated through crossover and mutation operations. The top-$N$ individuals are then selected to form the next generation. The evolutionary process continues in this round-robin fashion until a stopping criterion is met. Finally, the top-$k$ feasible designs are selected and passed to ground-truth evaluation.

---

**Algorithm 1** Constraint-aware evolutionary search

---

**Input:** CPC surrogate: $\hat{y}_f$ and $\hat{y}_p$, Population size: $N$, Parent population size: $S$, infeasible acceptance probability: $\lambda$, feasibility threshold: $\theta$, total generations: $T$, final ground-truth evaluation size: $k$.

**Output:** Best feasible solutions $\mathcal{B}$.
1: $y_p^* \leftarrow +\infty, \mathcal{B} \leftarrow \emptyset$
2: $\mathcal{P}^{(1)} \leftarrow$ randomly sampling.
3: Sort $\mathcal{P}^{(1)}$ in ascending order of predicted latency $\hat{y}_p(x)$
4: **for** $t \leftarrow 1 : T$ **do**
5:     $\mathcal{P}^{(t+1)} \leftarrow \emptyset$
6:     **for** every $x$ in $\mathcal{P}^{(t)}$ **do**
7:         **if** $\hat{y}_f(x) \geq \theta$ **then**
8:             Add $x$ to feasible archive: $\mathcal{B} \leftarrow \mathcal{B} \cup \{x\}$
9:             Add $x$ to next generation: $\mathcal{P}^{(t+1)} \leftarrow \mathcal{P}^{(t+1)} \cup \{x\}$
10:            Update best feasible solution $y_p^* \leftarrow \min(y_p^*, \hat{y}_p(x))$
11:         **else if** $\hat{y}_p(x) < y_p^*$ **and** rand() $< \lambda$ **then**
12:            Add promising infeasible $x$ to next generation: $\mathcal{P}^{(t+1)} \leftarrow \mathcal{P}^{(t+1)} \cup \{x\}$
13:         **end if**
14:         **if** $|\mathcal{P}^{(t+1)}| \geq S$ **or** $\mathcal{P}^{(t)} = \emptyset$ **then**
15:            **break**
16:         **end if**
17:     **end for**
18:     $\mathcal{C} \leftarrow$ apply crossover and mutation on $\mathcal{P}^{(t+1)}$
19:     $\mathcal{P}^{(t+1)} \leftarrow \mathcal{P}^{(t)} \cup \mathcal{C}$
20:     Sort $\mathcal{P}^{(t+1)}$ and $\mathcal{B}$ in ascending order of predicted latency $\hat{y}_p(x)$
21:     $\mathcal{P}^{(t+1)} \leftarrow \mathcal{P}^{(t+1)}[: N]$
22:     $\mathcal{B} \leftarrow \mathcal{B}[: k]$
23: **end for**

---

## C Additional Experiment Results and Experimental Details

### C.1 Details of Experimental settings of Zero-Shot

In this section, we introduce context vectors $c$ that summarize key properties of each dataflow, including tiling, clustering, and mapping decisions. Concretely, for a DL model represented by the dimension set $D = [K, C, R, S, X, Y, X', Y']$, each represents output channel, input channel, filter row, filter column, input row, input column, output row, and output column respectively (Kwon & et al., 2019). We extract binary features indicating whether each dimension is covered by a SpatialMap ($smp$) or TemporalMap ($tmp$) both before (pre) and after (post) clustering, yielding 32 features:

$$[\text{pre}_{smp}(K), \text{pre}_{tmp}(K), \cdots, \text{pre}_{smp}(Y'), \text{pre}_{tmp}(Y'), \text{post}_{smp}(K), \text{post}_{tmp}(K), \cdots, \text{post}_{smp}(Y'), \text{post}_{tmp}(Y')].$$

In addition, five global features are included to capture the presence of cluster operations and specific dimension tiles: $\text{has}_{\text{cluster}}$, $\text{has}_{\text{Ktile}}$, $\text{has}_{\text{Ctile}}$, $\text{has}_{X'}$, $\text{has}_{Y'}$. Together, these 37-dimensional context vectors $c \in \mathbb{R}^{37}$ describe the accelerator dataflow, allowing the surrogate to generalize across application domains.

In our experiments, CODA is trained on a set of train dataflows. For given train dataflows $q = 1, \ldots, Q$, each accelerator architecture in the offline dataset is represented by an input pair $(x, c_q)$. Here, $x$ is a specific accelerator design annotated with its latency and feasibility, while $c_q$ defines the different dataflow type. For test dataflow $q_t$, the learned surrogate $\hat{y}$ is then applied to test dataflows with distinct context vectors $\hat{y}(x, c_{q_t})$.

This allows us to achieve **zero-shot generalization**: once a contextual surrogate is learned, we can optimize it for a novel context vector to find the optimal accelerator for a previously unseen dataflow, without any labeled data for that specific dataflow. The surrogate predicts the feasibility and latency for candidate designs based on the context vector, allowing the optimizer to explore promising regions of the design space efficiently.

## C.2 Feasibility Classification Results on all applications

We evaluate the feasibility classification accuracy across all 10 cases studied in this paper. We consider two key aspects: 1) classification performance on previously unseen test sets, and 2) comparisons with other classical classifiers. The confusion matrices are further provided in Fig. 6, illustrating the reliability of CPC in distinguishing feasible from infeasible designs. Even in challenging cases such as case 9, CPC maintains a high F1-score. This strong feasibility classification lays the foundation for effective downstream performance prediction and optimization.

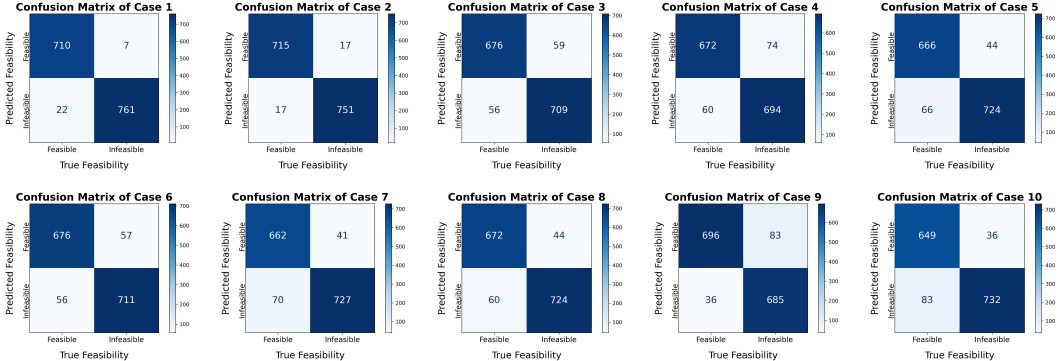

Figure 6: Confusion matrices across 10 cases.

To assess the classification performance, we compared the CPC network with several standard baseline classifiers, listed in Sec. 5 All baseline hyperparameters were tuned on the training dataset using four-fold cross-validation, and the final results are reported on the test dataset. Table 2 in Sec. 5.3 displays that Logistic Regression and Linear SVM achieve competitive yet moderate performance across most settings, with F1-scores typically around 0.85. Random Forest and LightGBM perform better than linear baselines in certain cases (Case 1, 2, 7 and 8), but their performance degrades significantly in other architectures. In contrast, our CPC network consistently outperforms all baselines across all architectures, achieving F1-scores above 0.90 in nearly all settings and as high as 0.98 on case 1. This demonstrates the effectiveness of out cascaded surrogate for constraint-performance prediction in capturing feasibility patterns and delivering robust prediction accuracy across diverse applications and dataflows. Overall, within the vast solution space, CPC network consistently delivers stable and superior performance, demonstrating its effectiveness for constraint-aware optimization tasks.

## C.3 Performance Prediction Results on all applications

We evaluate the Spearman's rank correlation coefficients between CPC network predictions and the ground-truth latency, which reflect the fidelity of the surrogate model. As shown in Table 7,

CPC network achieves consistently high correlations across all test sets, with an average Spearman coefficient of 0.99. Despite being trained on limited offline datasets, CPC network provides reliable performance predictions that effectively guide the evolutionary search toward better solutions without requiring costly simulator queries.

Table 7: Spearman's rank correlation coefficients results on test sets (between CPC network predictions and the ground-truth latency).

| Case 1 | Case 2 | Case 3 | Case 4 | Case 5 | Case 6 | Case 7 | Case 8 | Case 9 | Case 10 |
|--------|--------|--------|--------|--------|--------|--------|--------|--------|---------|
| 0.996  | 0.997  | 0.987  | 0.987  | 0.989  | 0.988  | 0.995  | 0.995  | 0.992  | 0.992   |

In detail, scatter plots were also provided illustrating the rank correlation between CPC network predictions and ground-truth latency values in Fig. 7. Each subplot corresponds to a specific accelerator case scenario, demonstrating the ability of CPC network to accurately preserve the performance ranking of unseen accelerator architectures. The near-perfect alignment in all cases visually confirms the high Spearman's rank correlation coefficients reported in Table 7, highlighting CPC network's exceptional fidelity in predicting relative performance trends. This ranking preservation is crucial for effectively guiding the evolutionary search toward optimal designs without requiring expensive simulator evaluations.

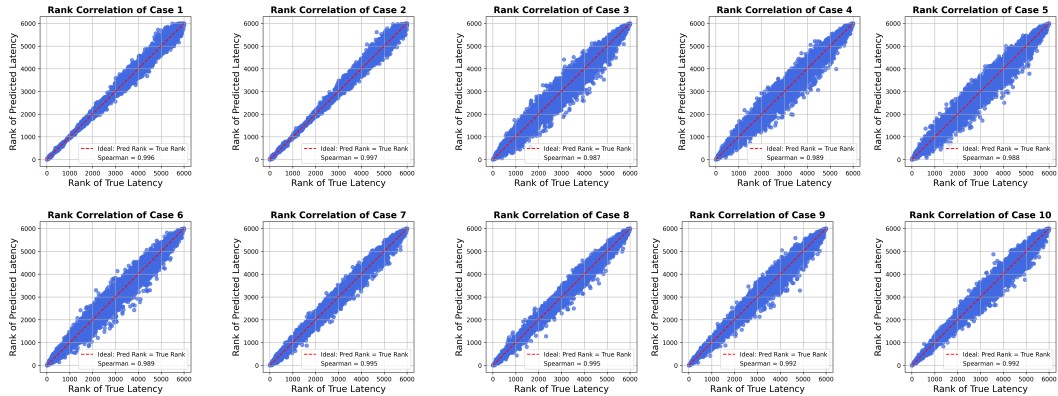

Figure 7: Scatter plots illustrating the rank correlation between CPC network predictions and ground-truth latency values across 10 cases.

### C.4 Details of Hyperparameter Analysis

In this section, two major hyperparameters were analysed. CODA exposes two key factors that most affect constraint-aware accelerator search: 1) the feasibility threshold $\theta$ used by the classifier to decide whether a candidate architecture is regarded as feasible or infeasible; and 2) the infeasible acceptance probability $\lambda$ used during optimization to explore beyond the currently predicted feasible set.

To validate the rationality of selecting $\theta = 0.5$, several Precision–Recall curves were plotted to compare the difference between the Best F1-score threshold in the test data set and the fixed threshold of 0.5. As shown in Fig. 8, the F1-scores under both thresholding strategies are nearly identical and consistently high across most cases. Although a minor discrepancy of 0.01 is observed in Mnas-Net–NVDLA and MobileNetV2–NVDLA, all other datasets achieve identical F1-scores up to two decimal places.

We further investigate the infeasible-acceptance probability threshold $\lambda$, examining how its selection influences the performance of our proposed model. To analyze the impact of this parameter, we vary $\lambda$ from 0.05 to 0.5 and evaluate the corresponding model behavior. Experimental results demonstrate that $\lambda = 0.1$ achieves the best performance in most cases, particularly for ResNet-ShiDianNao. Based on these findings, we select $\lambda = 0.1$ as the optimal value for this hyperparameter.

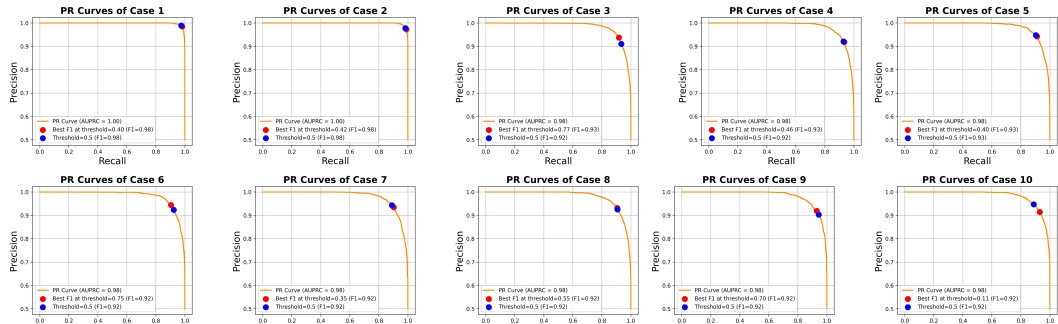

Figure 8: Precision–Recall Curves: Best F1 vs. Fixed Threshold (0.5): The red point marks the Best F1-score threshold on the test set. The blue point shows the F1-score achieved with a fixed threshold of 0.5.

Table 8: Model Performance with Varying Infeasible-Acceptance Probability $\lambda$. EXP Decay refers to Exponential Decay, which decreases as the number of generations increases. The optimal results are highlighted in bold.

| Case | $\lambda$ | | | | | |
|---|---|---|---|---|---|---|
| | 0.05 | 0.1 | 0.2 | 0.3 | 0.4 | 0.5 |
| Case 1 | 36.77 | 36.78 | **36.68** | **36.68** | **36.68** | 36.78 |
| Case 2 | **1.49** | **1.49** | **1.49** | **1.49** | **1.49** | **1.49** |
| Case 3 | 2.59 | **2.55** | 2.65 | 2.59 | 2.60 | 2.61 |
| Case 4 | **3.13** | **3.13** | 3.19 | 3.14 | 3.17 | **3.15** |
| Case 5 | 2.62 | 2.58 | **2.55** | 2.60 | 2.60 | 2.63 |
| Case 6 | 2.54 | 2.54 | 2.61 | 2.62 | 2.54 | **2.52** |
| Case 7 | 32.89 | 32.98 | 32.81 | 33.49 | 32.94 | **32.69** |
| Case 8 | 34.09 | **33.12** | 33.84 | 33.84 | 33.82 | 33.62 |
| Case 9 | 1.59 | **1.57** | **1.57** | 1.60 | 1.62 | 1.61 |
| Case 10 | 1.40 | 1.40 | 1.40 | 1.40 | **1.38** | 1.42 |
| **Geomean of CODA Improvement** | 1.0049 | 1.0000 | 1.0066 | 1.0099 | 1.0062 | 1.0075 |

## C.5 DETAILS OF ABLATION STUDY

To evaluate the contribution of key components in our proposed CODA, we conduct an ablation study by systematically removing or modifying specific elements. The following variants are examined:

- **CODA-w/o feasibility prediction** (*CODA-w/o FP*): This variant removes the feasibility classifier, so the CPC network produces only a single output focused on performance prediction. During evolutionary search, candidate solutions are selected solely based on this performance predictor. This variant is to evaluate the contribution of feasibility identification in guiding the search toward valid accelerator designs.

- **CODA-w/o cascaded encoders** (*CODA-w/o CE*): This variant simplifies the architecture by eliminating the cascaded encoders, using a single encoder that outputs both feasibility and performance predictions simultaneously. This variant is to assess the contribution of the cascaded encoder design in surrogate modeling.

- **CODA-w/o uncertainty weighting** (*CODA-w/o UW*): This variant replaces the uncertainty weighting mechanism with a fixed weight, in order to analyze the impact of learned weighting on prediction performance.

- **CODA-w/o infeasibility tolerance** (*CODA-w/o IT*): This variant disables the mechanism that allows the inclusion of a few infeasible designs during the search process, by setting the acceptance probability to $\lambda = 0$, thereby restricting the evolutionary search to feasible solutions only. This variant is to examine the benefit of leveraging infeasible designs during the search.

- **CODA-w/o Transformer** (*CODA-w/o TR*): This variant replaces the Transformer-based encoder with a standard Multi-Layer Perceptron (MLP) encoder of comparable parameter count. It is designed to evaluate the importance of modeling topological dependencies across accelerator layers using positional encoding and multi-head attention mechanisms.
- **CODA-w/o Evolutionary Search** (*CODA-w/o ES*): This variant replaces CODA's evolutionary search with Random Search while retaining the same pre-trained CPC surrogate for candidate evaluation. This ablation is designed to isolate and quantify the contribution of the evolutionary search algorithm to the overall optimization performance.

Table 4 summarizes the performance of each variant compared to the full CODA model. Ablation results demonstrate that the complete model consistently achieves competitive or superior performance across most hardware accelerator applications, confirming the value of each ablated component.

### C.6 EXTENSION TO MULTI-CONSTRAINT SCENARIOS

Our framework formulates feasibility as a binary classification problem. Algorithmically, this treats the feasible region as the logical intersection of all individual constraints, $feasible(x) = 1 \iff C_{area} \cap C_{power} \cap C_{bandwidth} \cap \cdots$. The CPC network abstracts away the complexity of balancing conflicting penalty terms by learning a single, unified feasibility boundary. While this increases the complexity of the decision boundary, the high capacity of our Transformer-based encoder is specifically suited to capture these complex, non-linear manifolds.

To prove this empirically, we conducted a new experiment incorporating Area, Power, and NoC bandwidth constraints. The new constraints were set at the median value of the dataset, creating a challenging optimization landscape with a significantly reduced feasible region. As shown in the 9, CODA maintains highly competitive results compared to both online and offline methods. These results confirm that CODA's feasibility modeling paradigm is not only conceptually natural but also practically effective in multi-constraint settings.

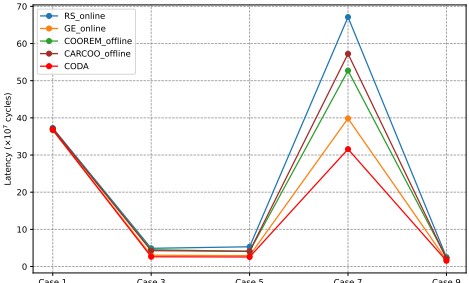 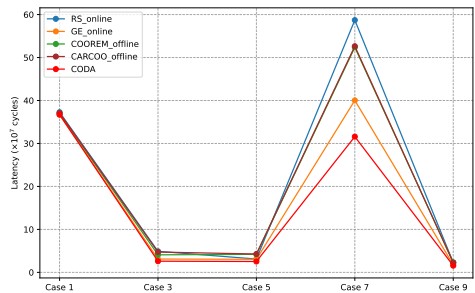

Figure 9: Optimization results (lower is better) comparison against online and offline optimization methods under multi-constraint scenarios. **Left**: Experiments incorporating both Area and Power constraints; **Right**: Experiments incorporating three constraints: Area, Power and NoC bandwidth.

## D QUALITATIVE ANALYSIS OF CPC CHALLENGES

In order to understand the practical strengths and limitations of our surrogate, we conduct a qualitative analysis of the CPC surrogate's error modes.

**Analysis of Optimization Trajectory and Feasibility Drift.** We first analyzed the behavior of CODA throughout the optimization process by tracking the performance of all populations and the final archive across generations, as well as the number of feasible solutions in the final archive under ground-truth evaluations. Results are shown in Figure 10. The performance of the best-found solution shows a consistent, monotonic improvement across generations, ultimately converging to a high-quality result. Critically, the optimization does not stall in a local optimum or prematurely converge due to strict constraints, demonstrating the effectiveness of the performance predictor. Moreover, the archive's best solutions (red curve) closely track the true best performance found in

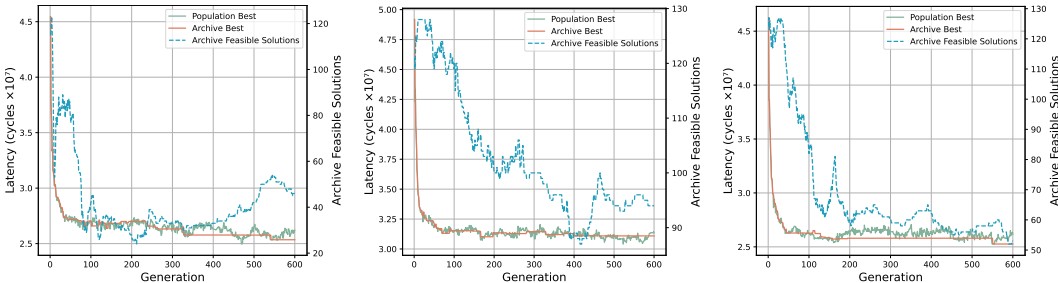

Figure 10: Performance and feasibility analysis across generations under ground-truth evaluations in Cases 3, 4 and 6. The left y-axis shows true performance, including both population best and archive best, while the right y-axis indicates the number of feasible solutions in the archive.

the population (green curve), indicating that the surrogate-guided archive maintains high-quality candidates that are strongly correlated with ground-truth performance.

For the feasibility classifier, we observe that the proportion of feasible solutions within the population decreased as the search progressed. This indicates that the evolutionary algorithm, driven by the performance predictor, progressively explores regions of the design space that are increasingly distant from the distribution of the original offline training data. The feasibility classifier, trained on this initial distribution, becomes less certain and makes more errors in these novel regions, a common challenge for models in offline optimization settings.

**Classification Error Analysis on the Final Population.** To precisely characterize the nature of CPC's prediction errors, we performed a detailed analysis of the final generation's population (500 designs) for a representative high-dimensional Case 6. The confusion matrix between CPC's predictions and the ground-truth is as shown in Table 9.

This analysis reveals two critical insights about the impact of prediction errors: 1)The model correctly identified all 218 truly feasible designs (True Positives) without any false negatives. This guarantees that every high-performance, manufacturable design is passed to the final performance predictor and has a chance to be selected for ground-truth evaluation. 2)Although a significant number of False Positives exist, their impact on the outcome is inherently mitigated by CODA's two-stage architecture. These misclassified designs, upon being passed to the performance predictor, would only enter the final candidate pool if they were also predicted to be top performers. The final ground-truth evaluation then acts as an ultimate verifier, filtering out any such designs that are, in reality, infeasible or poor performers.

Table 9: Confusion Matrix of Ground Truth vs Prediction

|  | Predicted Feasible | Predicted Infeasible |
|---|---|---|
| True Feasible | 218 (TP, 43.6%) | 0 (FN, 0%) |
| True Infeasible | 209 (FP, 41.8%) | 73 (TN, 14.6%) |

**Conclusion and Future Work.** In conclusion, the qualitative analysis confirms that the primary error mode is over-optimism in complex, under-sampled subspaces. However, these errors do not critically impact the optimizer's ability to find high-quality solutions. A feasibility classifier with a strong bias against false negatives, coupled with a final ground-truth verification step, ensures robustness and delivers reliable results, as evidenced by the performance improvements reported in the main experiments. These findings suggest that future work should focus on enhancing the model's sensitivity to subtle constraint violations.

## E    USE OF LLMS IN THE PAPER

In preparing this paper, we employed large language models (LLMs) as a general-purpose writing assistant. The LLMs were used to aid in phrasing, improve readability, and polish grammar.

For instance, in Sec. 5.1, we use ChatGPT to polish our sentences:

**Before:** Overall, these results underscore CODA's ability to maintain high feasible solution ratio and deliver reliable latency performance, which emphasizes its practical utility in accelerator design problems.

**After:** Overall, these results highlight CODA's ability to simultaneously maintain a high feasible solution ratio and deliver competitive latency performance, emphasizing its practical utility in offline accelerator design.

We emphasize that LLMs were employed solely for writing purposes. All technical ideas, experimental design, analysis, and conclusions were conceived and developed entirely by the authors. The LLM was not involved in research ideation, methodology development, or result interpretation.

