# OpenReview forum: "CODA: Learning to Guide Constraint-Aware Optimization for Hardware Accelerators"
_ICLR.cc/2026/Conference — ICLR 2026 Conference Withdrawn Submission_

### Official Review · Reviewer_TbNx · 2025-10-25

**Soundness:** 2
**Presentation:** 3
**Contribution:** 2
**Rating:** 4
**Confidence:** 3

**Summary:**

The paper introduces CODA, a constrained offline optimization framework for DNN hardware accelerator design. It assumes access to an offline dataset of simulated designs and focuses on minimizing inference latency while satisfying feasibility (primarily area constraints). Latency is evaluated by the analytical model MAESTRO; feasibility is treated as a binary indicator.

The core is a Constrained‑Performance Cascade (CPC) surrogate: a feasibility classifier (first gate) and a latency predictor (second stage), trained jointly with uncertainty‑weighted multi‑task learning to cope with data sparsity. Search uses a constraint‑aware evolutionary algorithm where predicted infeasible candidates are filtered, but a small fraction can be retained to preserve diversity.

**Strengths:**

* Cascading feasibility and performance with uncertainty‑weighted multi‑task learning is an intuitive and sensible adaptation to avoid wasting search on invalid points while still learning a performance‑useful representation from limited data.
* Combining it with a diversity‑preserving evolutionary loop is practical. Retaining a small fraction of predicted‑infeasible designs is a simple yet effective diversity knob as analyzed in the paper.
* The empirical gains are notable on the chosen benchmarks.

**Weaknesses:**

* The formulation Eq. (1) assumes a per‑layer pair and evaluates latency/feasibility on that basis. This implies that a one‑layer–to–PE‑resource attribution that does not capture:
    1. Mapping of multiple layers per PE array,
    2. Tiling of a single layer across many PEs and time steps, and
    3. Temporal multiplexing across layers/an operator fusion pipeline.
* None of this utilization/partitioning complexity is explicit in the representation, yet it critically affects latency and feasibility in real accelerators.
* The paper explicitly focuses on area as the feasibility condition, noting that infeasibility could also result from memory/power or mapping failures, but these are not modeled in the experiments; NoC connectivity, bandwidth, off‑chip DRAM traffic, on‑chip SRAM bank conflicts, and load balance do not appear in the feasibility labels or the objective. This limits external validity for practical accelerator design.
* Latency is measured by MAESTRO (which is analytical), not cycle‑accurate simulation or silicon measurements. Conclusions depend on its fidelity for the chosen dataflows. In comparative evaluations, CODA enjoys 6,000 pre‑labeled designs per case offline, whereas online baselines are limited to the same query budget but without that offline corpus. This raises fairness concerns.
* The zero‑shot transfer is across three canonical dataflows with a hand‑crafted 37‑D context. It is unclear whether this generalizes to novel dataflows (e.g., different unrolling axes, GEMM‑centric systolic arrays, sparsity‑aware engines) or new hardware knobs.

**Questions:**

* On Eq. (1), how does CODA account for (a) multi‑layer sharing of a PE array, (b) spatial/temporal tiling of a single layer over many PEs, and (c) load balance across PEs? If these are only in the dataflow context vector, can you show that the context features uniquely determine the mapping‑relevant degrees of freedom for latency and feasibility?
* Can feasible(x) be extended to include other constraints such as NoC/SRAM/DRAM bandwidth and port/banking conflicts?
* Could you test zero‑shot on a new dataflow family (e.g., GEMM‑systolic arrays with array size/tiling knobs), or inject sparsity/activation compression features into the context and measure transfer?
* Can CODA handle multi‑head prediction (optimize latency, energy, and area jointly) with Pareto‑front search?

---

> ### Author Response · Authors · 2025-12-03
> **Response to Reviewer #TbNx (part 1/2)**
>
> We are particularly grateful for their recognition of the intuitive and sensible design of our cascaded feasibility-performance surrogate, the practical integration with a diversity-preserving evolutionary search loop, and the notable empirical gains. Below, we provide a detailed response to the remaining concerns raised.
>
> **[W1 & W2 & Q1  representation granularity & hardware mapping realism]**
>
> We agree that factors like tiling and multiplexing are critical. However, we clarify that Eq. (1) defines the architectural search space, while the complex execution logic is handled by the dataflow and the Analytical Model, which our CPC model learns to approximate.
>
> 1. **Separation of Concerns:** Eq. (1) represents the design variables available to the architect. It is not meant to explicitly encode the schedule. In our framework, the mapping strategies, such as multi-layer serialization, spatial/temporal tiling, and data reuse patterns, are determined by the dataflow and the DNN layers’ dimensions. These are inputs to the evaluation, not variables to be searched
> 2. **Implicit Learning of Complexity**: Although Eq. (1) appears simple, CODA captures the impact of (a) sharing, (b) tiling, and (c) load balancing through Implicit Learning: 1) **ground-truth supervision:** The surrogate is trained on latencies generated by MAESTRO, which *does* explicitly compute tiling, spatial mapping, and reuse based on the hardware constraints provided. 2) **Dataflow Context:** As detailed in Appendix C.1, we introduce a context vector that explicitly encodes mapping decisions. **3) Surrogate Capability**: By conditioning the prediction on both the hardware resources and the mapping context, the CPC surrogate learns the complex function: $f(\text{Resources}, \text{Mapping}) \rightarrow \text{Latency}$.  The high rank correlation of 0.99 (Section 5.3) confirms that the model successfully captures how resource changes affect latency under these complex mapping rules.
> 3. **Sufficiency of Context Features:** The reviewer asks if context features determine mapping degrees of freedom. Yes, for a fixed dataflow, the mapping policy is deterministic given the hardware resources and layer shape. The context features uniquely identify these dataflow constraints. Therefore, the combination of ”input architecture + context vector” provides sufficient information for the surrogate to predict the performance impact of tiling and scheduling without needing to explicitly search for them.
>
> **[W3 & Q2 other infeasibility causes/constraints]**
>
> We confirm that the CODA framework can be naturally extended to accommodate these more complex constraints, and we provide new empirical evidence to validate its robustness. The core architecture of CODA is the cascaded surrogate model (CPC) and constraint-aware evolutionary search. It is designed to be agnostic to the specific definition of feasibility. Its objective is to learn the mapping from design parameters to feasibility and performance, regardless of how feasibility is defined.
>
> To validate this capability, we conducted experiments incorporating Area, Power constraints, and NoC Bandwidth, creating a challenging optimization landscape with a reduced feasible region. The results (Latency, lower is better) demonstrate that CODA maintains strong performance under these three constraints, consistently outperforming online methods (RS and GE) and offline methods (COOREM and CARCOO). The framework's performance remains robust when additional practical constraints are introduced, confirming its applicability to real-world accelerator design problems.
> | Case | CODA | RS | GE | COOREM | CARCOO |
> | --- | --- | --- | --- | --- | --- |
> | 1 | **36.71** | 37.28 | 36.72 | 37.03 | 37.08 |
> | 3 | **2.60** | 4.90 | 3.10 | 4.08 | 4.73 |
> | 5 | **2.53** | 4.73 | 3.04 | 4.20 | 4.27 |
> | 7 | **31.59** | 58.71 | 40.03 | 52.38 | 52.64 |
> | 9 | **1.56** | 2.28 | 1.75 | 2.31 | 2.25 |
>
> **[W4 experiment fairness & Simulation Fidelity]**
>
> We clarify that our experimental design enforces strict budget parity. If anything, the setup fundamentally favors the online baselines:
>
> - **Strict Budget Parity:** Every method is limited to exactly **6,128 ground-truth evaluations**. Online baselines utilize the entire budget for active, feedback-driven optimization. In contrast, CODA is restricted to 6,000 static samples for training and only 128 for final validation. This setup places CODA at a disadvantage, as it lacks the ability to actively query the simulator during the search, unlike online methods.
> - **Uniform Fidelity:** While MAESTRO is an analytical model, it serves as the consistent ground-truth oracle for **all** methods in this study. Any potential modeling constraints apply uniformly to all algorithms, ensuring that the reported relative performance gains are unbiased and valid.

---

> ### Author Response · Authors · 2025-12-03
> **Response to Reviewer #TbNx (part 2/2)**
>
> **[W5 & Q3 generalization to novel dataflows]**
>
> We clarify a key conceptual distinction: "Dataflow" is fundamentally defined by the choice of unrolling axes and mapping strategies (i.e., which loops are spatially or temporally mapped), rather than the specific computation kernel (like GEMM). Based on this, our framework is inherently generalizable: The 37-D context vector (Appendix C.1) explicitly encodes the fundamental primitives of dataflow: SpatialMap and TemporalMap across varying tensor dimensions.
> In Table 1, we show zero-shot transfer between NVDLA, ShiDianNao, and Eyeriss dataflows, which embody fundamentally different unrolling axes, thereby demonstrating our model's capability to generalize across these core mapping decisions.
>
> In this context, we note that MAESTRO’s analytical model is agnostic to the computational kernel (e.g., GEMM or convolution). It focuses on the performance implications of data movement and computation scheduling across the memory hierarchy and processing elements. In this context, 'GEMM-centric systolic arrays' do not represent a novel class of dataflow, but rather implement specific instances of loop unrolling and mapping strategies. Since our context vector already captures the combinatorial space of spatial/temporal mappings and clustering strategies, our CPC model can readily handle such GEMM-style mappings without structural modification.
>
> We currently do not support sparsity-aware engines primarily because the MAESTRO used in our work does not natively model the dynamic execution patterns and load imbalance inherent in sparse computations. Incorporating sparsity and activation compression features presents a significant but worthwhile challenge. We envision this would require a dual extension: extending the context vector with features describing the sparsity pattern, compression ratio, and integrating a sparsity-aware performance model.
>
> **[Q4 multi-head prediction]**
>
> We confirm that CODA is architecturally and algorithmically capable of handling multi-objective optimization with minimal modification:
>
> - **Architectural Readiness:** The CPC network is inherently designed as a multi-head predictor. Extending this to include Power or Area heads is a direct expansion of Eq.(6). The shared representation learning (Section 4.1) is particularly well-suited for this, as it naturally captures the correlations between competing hardware objectives.
> - **Pareto Search Integration:** The evolutionary search module (Section 4.2) serves as a flexible optimizer. Replacing the current single-objective ranking with non-dominated sorting (NSGA-II) would seamlessly enable pareto-front exploration without altering the core surrogate mechanism.
> - **Current Focus:** While multi-objective optimization is a logical extension, this work prioritizes the fundamental bottleneck of feasibility identification in sparse data regimes. We therefore view multi-head prediction and Pareto-front search as a highly promising and logical direction for future work.

---

### Official Review · Reviewer_w4kV · 2025-10-31

**Soundness:** 3
**Presentation:** 3
**Contribution:** 3
**Rating:** 6
**Confidence:** 3

**Summary:**

This paper introduces CODA, a framework for constraint-aware, surrogate-guided optimization of hardware accelerator designs from offline data. CODA leverages a cascaded surrogate neural architecture (CPC network) to first classify design feasibility (constraint satisfaction) then predict performance, and trains with an uncertainty-weighted dual-task learning approach to combat data sparsity. During search, a constraint-aware evolutionary algorithm exploits the surrogate for population filtering and ranking, balancing retention of feasible and informative infeasible solutions for exploration. Empirical results on several accelerator design cases demonstrate that CODA outperforms both online and leading offline optimization baselines in terms of performance and feasible solution ratio, and generalizes across unseen dataflows in zero-shot settings.

**Strengths:**

1. The paper addresses an important problem of hardware accelerator design optimization, which has practical significance in computer architecture and hardware design.

2. The experiments are thorough, covering multiple hardware accelerator design tasks (including AlexNet, MnasNet, MobileNetV2, ResNet50, and ShuffleNetV2). And the results demonstrate the effectiveness of CODA.

**Weaknesses:**

1. Extension to multi-constraint scenarios. CODA focuses almost exclusively on area constraints. While this is a practical starting point, most hardware accelerator design problems are simultaneously subject to multiple constraints such as power, memory bandwidth, etc. The extension to multi-constraint scenarios is claimed to be “natural”, but this remains unproven either empirically or algorithmically. Could the authors provide more discussion on the potential issues this may cause and possible solutions to address these issues?

2. Limited impact of some components. Certain parts of the method do not seem to contribute significantly. For example, why does CODA-w/o CE show only marginal improvement (0.0027)? Does this suggest that these components are not essential contributions?

3. Lack of qualitative analysis on failures. While Figures 5–7 show aggregate predictive correlations, the paper does not present qualitative examples (e.g., design configurations that CPC misclassifies or ranks incorrectly) that could help practitioners better understand the strengths and limitations of CPC. Could the authors provide such analyses?

4. The mysterious CPC network architecture. Overall, the design is good. However, could the authors elaborate more in the main text on the architectural design and underlying rationale of the CPC surrogate? For instance, what is the exact form of the input “solution”? How do positional encoding and multi-head attention function work in this context? How is the output of the Validation Encoder connected to the Evaluation Encoder?

5. The related work section does not compare this paper with other studies but merely lists existing works. How do the authors position this work among others?

6. Some typos: line 232 multi-tak. line 239 te.

**Questions:**

1. During the Offline Optimizing process, how much do different search algorithms (e.g., random search with filtering) affect the results? I understand that the baseline compares with online random search, but I think this is not equivalent to replacing the optimizing module in CODA with random search, right?

2. What happens if the final solution is infeasible? Moreover, since CODA only achieves around 60% feasibility, does this lead to significant computational waste?

---

> ### Author Response · Authors · 2025-12-03
> **Response to Reviewer #w4kV (part 1/3)**
>
> We are greatly appreciative of their recognition of the practical significance of the accelerator design optimization problem and the thoroughness of our experimental evaluation across multiple design tasks. Below, we provide a detailed response to the remaining concerns raised.
>
> **[W1 extension to multi-constraint scenarios]**
>
> We clarify that extending CODA to multi-constraint settings is **algorithmically straightforward,** and we provide new empirical evidence to validate its robustness.
>
> - **Algorithmic Generalization:** Our framework formulates feasibility as a binary classification problem. Algorithmically, this treats the feasible region as the logical intersection of all individual constraints, $feasible(x)=1 \iff C_{area} \cap C_{power} \cdots$. The CPC network abstracts away the complexity of balancing conflicting penalty terms by learning a single, unified feasibility boundary. While this increases the complexity of the decision boundary, the high capacity of our Transformer-based encoder is specifically suited to capture these complex, non-linear manifolds.
> - **Experimental Validation:** To prove this empirically, we conducted a new experiment incorporating both Area and Power constraints. The power constraint was set strictly at the median value of the dataset, creating a challenging optimization landscape with a significantly reduced feasible region. As shown in the table below (Latency, lower is better), CODA maintains highly competitive results compared to both online methods (RS and GE) and offline methods (COOREM and CARCOO). These results confirm that CODA's feasibility modeling paradigm is not only conceptually natural but also practically effective in multi-constraint settings.
>
> | Case | CODA | RS | GE | COOREM | CARCOO |
> | --- | --- | --- | --- | --- | --- |
> | 1 | 36.76 | 37.28 | **36.72** | 37.03 | 37.08 |
> | 3 | **2.62** | 4.90 | 3.04 | 4.53 | 4.26 |
> | 5 | **2.56** | 5.31 | 2.92 | 4.07 | 4.10 |
> | 7 | **31.58** | 67.11 | 39.84 | 52.71 | 57.23 |
> | 9 | **1.55** | 2.50 | 1.73 | 2.30 | 2.17 |
>
> **[W2 impact of cascaded encoder]**
>
> While the average numerical improvement appears modest, the Cascaded Encoder functions as a critical structural refinement rather than the primary performance engine. By explicitly learning simultaneously of latency predictions and feasibility features, it enforces a logical dependency essential for disentangling these factors in high-dimensional spaces. The aggregate score is diluted by simple tasks; however, in complex architectures (Cases 3 and 8), the component delivers consistent gains of ~2%, which is a meaningful margin in hardware optimization. Given that this integration imposes negligible computational overhead while enhancing robustness in **8 out of 10 cases**, it represents a cost-effective architectural constraint that ensures stability in complex design spaces.
>
> **[W3 qualitative analysis on prediction errors]**
> We thank the reviewer for highlighting the need for a qualitative analysis of CPC's error modes. We have added a dedicated section in **Appendix D** to address this. Our findings show that:
>
> 1. **Optimization Trajectory:** The best-found solutions improve steadily across generation, demonstrating the robustness of CPC’s performance predictor. Moreover, the archive’s best solutions closely track the true best performance found in the population, indicating that the surrogate-guided archive maintains high-quality candidates that are strongly correlated with ground-truth performance.
> 2. **Feasibility Drift:** As the search progresses, the proportion of feasible solutions within the population decreases, indicating that CPC explores regions increasingly distant from the original training distribution.
> 3. **Classification Errors**: As the search progresses, the proportion of feasible solutions within the population decreases, indicating that CPC explores regions increasingly distant from the original training distribution.
>
> In conclusion, the qualitative analysis confirms that the primary error mode of our CPC model does not critically impact the optimizer's ability to find high-quality, feasible solutions. Future work will focus on enhancing the model’s sensitivity to subtle constraint violations, for example by refining feature representations in critical regions or applying targeted sampling strategies.

---

> ### Author Response · Authors · 2025-12-03
> **Response to Reviewer #w4kV (part 2/3)**
>
> **[W4 CPC network architecture details]**
>
> We clarify the specific mechanisms and their underlying rationale details below:
>
> 1. **Input representation:** The input to the CPC network is a candidate accelerator design, which is represented as a sequence with length $n$, (the target DNN model $n$ layers) consisting of discrete tuples. These tuples ($PEs_i$, $Buffers_i$) are projected into a continuous, learnable embedding space, allowing the model to capture the semantic similarities between different resource levels.
> 2. **Rationale of Transformer components: 1) Positional encoding:** In accelerator design, the topological order is critical because data flows sequentially from Layer $i$ to Layer $i+1$. Positional encoding injects this structural prior, allowing the model to distinguish between identical configurations placed at different stages of the pipeline. **2) Multi-head attention**: Hardware performance is governed by global dependencies. The self-attention mechanism enables the model to capture these non-local interactions. It allows the representation of a specific layer to dynamically "attend" to other layers, learning complex resource coupling effects such as inter-layer data reuse and pipeline stalling. The rationale for employing the Transformer structure is strongly supported by our ablation studies. (See results in Table 4)
> 3. **Cascaded connection:** The connection between the two encoders is a direct feature reuse mechanism. The Validation Encoder processes the raw sequence to produce a high-level latent representation, $h_{val}$, which primarily captures structural feasibility features. Instead of restarting from scratch, the Evaluation Encoder takes $h_{val}$ as its input. The Evaluation Encoder refines this shared representation, focusing on isolating features that are specifically predictive of performance.
>
> **[W5 related work & positioning]**
>
> We apologize for the lack of explicit comparative analysis in the related work section due to space limitations. We position CODA as a specialized framework that bridges the gap between **general offline optimization** and **domain-specific hardware design**, addressing limitations that prior works tackle only partially. Specifically:
>
> 1. **Vs. Online Methods**. Unlike online approaches that rely on expensive iterative simulations, CODA is positioned as a data-efficient offline solution. We eliminate the need for active simulator queries during the search, enabling scalable hardware design space exploration with tiny simulation cost at the inference process.
> 2. **Vs. General Offline BBO**. These general offline algorithms employ trivial, domain-agnostic methods for constraint handling. In contrast, by utilizing a CPC model, our CODA captures the topological dependencies of accelerator layers, which is a critical structural nuance that general-purpose BBO methods fail to model, leading to their poor performance in high-dimensional cases (as shown in Fig. 4).
> 3. **Vs. SOTA Offline method**. PRIME[1]  is the closest existing work. We advance beyond PRIME by addressing its primary weakness: the handling of strict feasibility constraints. While PRIME relies on conservative objective estimation, CODA introduces the constraint-performance cascade (CPC) and dual-task learning. This allows for a more fine-grained disentanglement of feasibility and latency features, ensuring that the optimizer is not just finding "good" designs, but "manufacturable" ones, which is evidenced by our significantly higher feasible solution ratios.
>
> We have revised **Section 2** to explicitly articulate these comparative distinctions rather than simply listing the methods.
>
> [1]Yazdanbakhsh, Kumar et al. “Data-Driven Offline Optimization for Architecting Hardware Accelerators”. 2022 ICLR
>
> **[Q1 Impact of Search Algorithm]**
>
> We conducted an ablation study: replacing CODA’s evolutionary search with Random Search (RS) while using the exact same pre-trained CPC surrogate. The results (Latency, lower is better) are summarized in **Section 5.4, Table 4**. In low-dimensional spaces (cases 1–2), RS performs comparably to CODA. The search space is small enough that random sampling can adequately cover the feasible manifold.
> In high-dimensional problems (cases 3–10), CODA consistently outperforms RS by significant margins. This confirms that in combinatorially large spaces, random sampling fails to locate the sparse high-performance regions. Unlike "stateless" random search, our evolutionary strategy maintains a population memory. It leverages the ranking guidance from the CPC to perform directed exploration. Furthermore, the random search with filtering cannot exploit the $\lambda$-acceptance mechanism effectively, meaning it cannot traverse infeasible regions to bridge disjoint feasible islands, which is critical for escaping local optima in complex landscapes.

---

> ### Author Response · Authors · 2025-12-03
> **Response to Reviewer #w4kV (part 3/3)**
>
> **[Q2 final solution feasibility & computational waste]**
>
> We confirm that CODA guarantees a valid final output and that the ~60% feasibility ratio represents high efficiency rather than waste:
>
> - **Feasibility Assurance:** The "final solution" is never an unchecked prediction. CODA outputs an archive of 128 candidates, all of which undergo ground-truth validation (using MAESTRO) at the very end. The final result is selected strictly from the verified feasible designs within this pool.
> - **Context of Sparsity:** In high-dimensional hardware design spaces, the feasible region is naturally extremely sparse. Achieving a ~60% feasibility ratio (significantly higher than baselines in Fig. 4) confirms that the CPC surrogate effectively concentrates the search into valid regions, filtering out the vast majority of invalid designs.
> - **Negligible Overhead:** Validating the ~40% infeasible candidates in the final step involves only ~50 wasted simulations. This cost is negligible compared to online optimization methods, which typically waste thousands of simulations on infeasible designs during the search process. Therefore, CODA significantly reduces total computational waste.

---

### Official Review · Reviewer_mCrF · 2025-11-01

**Soundness:** 2
**Presentation:** 3
**Contribution:** 2
**Rating:** 2
**Confidence:** 4

**Summary:**

This paper discusses an important problem in hardware accelerator design and DSE. It presents a framework for constraint-aware offline optimization of hardware accelerators which can predict both feasibility and performance. The surrogate model introduced in this paper incorporates uncertainty-weighted multitask learning to address data sparsity, and the evolutionary optimizer uses feasibility filtering to balance exploration and exploitation. It has an extensive experimental setup and claims to be performing better than online methods (1.11x) and SOTA offline methods (1.42x).

**Strengths:**

1. The problem discussed in the paper is timely and an important problem, especially in the current era of novel accelerators. And, it follows up with a good motivation with 4 clear reasons.
2. The surrogate that predicts both the feasibility and performance is an interesting choice and a direction worth exploring in this landscape.
3. Although there are limitations (explained below) in terms of the experiments, the paper has a sufficiently enough number og baselines to be compared with. Overall, a good experimental setup.
4. The paper claims to have better performance compared to both online and SOTA offline methods.
5. The paper is written well and easy to understand.

**Weaknesses:**

1. I think this paper might fall under "applications of ML" rather than "optimizations".
2. The model for the accelerators does not seem realistic. I could not find more information about the implementation of these accelerators or whether they are synthesizable, etc. I think this needs to go beyond an analytical model.
3.  The selection of the dataset sizes and choices is random and not well justified.
4. Paper introduces the difficulties in collecting data for the accelerators, which is indeed a very critical problem that needs to be addressed. However, it does not discuss about the time it took for their data collection effort and overheads.
5. It is questionable whether this solution is scalable beyond NVDLA, ShiDianNao and Eyeriss. If we look into the novel accelerator development landscape, the dataflow has become more complex.
6. It seems this approach relies a lot on the feasibility classifier. That might prevent it from reaching the global optimal.

**Questions:**

1. How are these accelerators generated? What is the end result here? Do we get the HDL implementation of these? Are these accelerators discussed here cycle-accurate? Are these synthesizable?
2. Can the introduced techniques in the CODA scale be applied in real-world scenarios where high dimensional hardware codesign problems involve memory hierarchy or NoC parameters?
2. I'm concerned about the size of the training dataset. Does this cover the search space well? The search space can be exponentially large. How did you come up with the number 6000 architectures for training? Can you also provide more context about how long it took to collect data?
3. How sensitive is this technique when it comes to the architecture of the encoders? Like Transformers and MLPs?
4. How does CODA handle misclassified feasible points? Could it reject globally optimal designs if the feasibility classifier errors?

---

> ### Author Response · Authors · 2025-12-03
> **Response to Reviewer #mCrF (part 1/3)**
>
> We are greatly appreciative of the novelty of our proposed surrogate model, the comprehensive nature of our experimental setup, and the overall clarity of the paper. Below, we provide a detailed response to the limitations and concerns raised.
>
> **[W1 paper scope]**
>
> We respectfully argue that our work contributes to the Machine Learning for Optimization domain rather than being a mere application. While we address hardware acceleration, we treat it as a representative instance of constrained black-box optimization over discrete, combinatorial spaces.
> Specifically, our contributions are algorithmic: (1) we propose a constraint-performance cascade network, a robust mechanism to explicitly model feasibility and avoid costly evaluations of invalid designs; (2) we develop a principled constraint-aware evolutionary search algorithm, which intelligently balances exploration and exploitation by selectively retaining promising yet infeasible candidates. Experimental results demonstrate CODA's effectiveness in solving complex, high-dimensional optimization problems under limited evaluation budgets, providing a methodology for data-efficient constrained optimization.
>
> **[W2 & Q1 hardware modeling realism & output definition]**
>
> We explicitly focus on early-stage hardware design space exploration, where evaluating millions of configurations requires a trade-off between speed and fidelity.
>
> - **Widely-used Practice Platform (Addressing W2)**: While we agree that analytical models are not perfectly "realistic" compared to silicon, using MAESTRO is the widely-used standard practice in this field (PRIME [1], ConfuciuX [2], GAMMA [3]). Running cycle-accurate simulation or synthesis for the vast search space required by learning-based optimization is computationally prohibitive. MAESTRO provides the necessary throughput for training while maintaining high fidelity (validated to have **<4% error** compared to RTL simulations) [4]. We emphasize that **all** baselines and our method are evaluated using the exact same ground-truth oracle (MAESTRO). Therefore, any modeling discrepancy affects all methods equally, ensuring that the reported relative performance gains are fair and valid.
> - **Definition of Output (Addressing Q1):**
>     1. **Nature of Output:** The output of CODA is a set of **optimized architectural configurations** (PE allocation, buffer sizing per layer) in the situation of a selected DNN model and dataflow, *not* raw HDL code. These configurations serve as high-quality blueprints for hardware engineers to synthesizable implement.
>     2. **Simulation Fidelity:** The accelerators discussed are evaluated using the analytical model MAESTRO, which is not cycle-accurate but provides high-fidelity performance estimations sufficient for ranking and optimization[5].
>
> **[Q2 scalability to high-dimensional codesign problems]**
>
> We thank the reviewer for the insightful question. We would like to clarify that CODA can indeed be extended to scenarios where memory hierarchy and NoC parameters are part of the design space. In addition to tuning the number of PEs and local buffer capacity, other parameters such as global buffer size, NoC bandwidth constraints, and NoC hop counts can be controlled in MAESTRO. Including these parameters increases the dimensionality of the design space, which does make the search problem more challenging. Although memory hierarchy and NoC parameters could be treated as independent design variables, our current work focuses on optimizing hardware resources to identify better accelerator configurations, following prior work[2-3].
>
> Our experiments demonstrate that CODA outperforms existing baselines in optimizing accelerator configurations with the currently selected design parameters. Incorporating additional memory hierarchy and NoC parameters as independent features is a natural extension and will be explored as future work, potentially enabling CODA to handle even more complex design spaces while still providing efficient performance prediction.

---

> ### Author Response · Authors · 2025-12-03
> **Response to Reviewer #mCrF (part 2/3)**
>
> **[W3 & W4 & Q3 selection of the dataset sizes & difficulties in collecting data]**
>
> We justified our choice of 6000 samples through sensitivity analysis and efficiency trade-offs. We conducted an ablation study by training and evaluating CODA with varying offline dataset sizes: 1,000, 3,000, 6,000, and 9,000. As shown in **Section 5.4, Table 3**, 6,000 samples strike the optimal balance.  In high-dimensional tasks, small datasets lead to an under-trained CPC model that fails to guide the search to any feasible solution. Increasing to 9000 yield only marginal gains (with the best improvement being only 0.18 in Case 7), but increases data collection costs by 50%.
>
> Based on our established literature and our experiments, cycle-accurate RTL simulation requires **7.2-28.8 hours** per simulation[4-5]. Even with analytical models like MAESTRO, each 112-dimensional evaluation takes about **2 seconds**. This underscores the significance of our CPC model, which reduces evaluation time to just about **0.2 seconds** - a 10× speedup over MAESTRO. This efficient surrogate modeling is precisely what enables CODA to perform comprehensive design space exploration that would otherwise be computationally prohibitive. The "cheap" evaluation provided by our approach is an essential enabler for tackling the enormous optimization spaces in accelerator design.
>
> **[W5 scalability to new dataflows]**
>
> We would like to clarify that the core strength of our method lies in its fundamental, data-centric representation of dataflows, which is not limited to the three canonical examples used in our initial evaluation. Our approach remains scalable to accelerators with more complex dataflows for three key reasons:
>
> 1. **Generalizability of the context representation.** Our 37-D context vector is not a fixed set of features for three specific dataflows. Instead, it is designed to encode the fundamental primitives of data movement and scheduling, and data movement order across key tensor dimensions (e.g., K, C, R, S, X, Y). These primitives are the building blocks for describing virtually any dataflow, whether simple or complex. CODA does not learn mapping decisions directly from a raw architectural description. The model's primary goal is to learn the mapping between architectural resource allocations (PEs, Buffers) and their performance outcomes under a given dataflow strategy, which is fully specified by the context vector.
> 2. **Data-driven scalability:**  The scalability of our CPC model is constrained not by its architectural design but by the diversity and complexity of the dataflows present in the training data. The Transformer-based encoder is renowned for its capacity to model complex, structured relationships. To scale to novel, complex dataflows, the required step is not a change to the CODA framework itself, but the inclusion of examples of these complex dataflows in the training set. Once trained on a sufficiently diverse set of dataflow descriptors, the model can generalize to new, unseen combinations of the same fundamental mapping primitives—even those constituting "novel" dataflows.
> 3. **Empirical evidence of generalization:** Our Zero-Shot generalization experiments provide empirical confirmation(Section 5.2). CODA successfully extracted meaningful patterns from the dataflow context vector of unseen accelerator configurations and identified high-performing designs. This demonstrates that our model does not merely memorize specific dataflows but learns the underlying performance principles of how mapping decisions affect latency and feasibility. This foundational understanding is the key to scaling to more complex designs.
>
> **[Q4 encoder sensitivity]**
>
> Our investigation confirms that the encoder architecture is a decisive factor for performance, particularly in high-dimensional design spaces. Accelerator configurations are not independent feature vectors but structured sequences corresponding to DNN layers. Resource allocation in one layer significantly impacts data reuse opportunities in others. The Transformer’s self-attention mechanism is theoretically necessary to capture these long-range, topological dependencies. In contrast, MLPs treat inputs as flat, unordered sets, failing to model the sequential interactions required for complex designs. To validate this, we compared our Transformer-based encoder against a standard MLP encoder. As shown in **Section 5.4, Table 4**, while *CODA-w/o TR*, a variant in which the Transformer-based encoder is replaced with a standard MLP encoder, perform adequately on low-dimensional tasks (Cases 1-2), it fail catastrophically on high-dimensional problems (Cases 3-10), where the search space is exponentially larger. The Transformer architecture is not a minor heuristic but a prerequisite for scalability. It enables CODA to navigate high-dimensional spaces where simple MLPs fail to grasp the structural constraints.

---

> ### Author Response · Authors · 2025-12-03
> **Response to Reviewer #mCrF (part 3/3)**
>
> **[W6 & Q5 robustness to feasibility misclassification]**
>
> We acknowledge that classifier errors are inevitable. However, CODA treats the feasibility classifier as a "soft guidance" rather than a hard bottleneck, employing specific mechanisms to prevent the rejection of global optima:
>
> 1. **Stochastic Recovery via Infeasible Acceptance:** To handle false negatives (feasible points misclassified as infeasible), we incorporate a stochastic tunneling mechanism. As detailed in Section 4.2, candidates predicted as infeasible are not discarded immediately; they are retained for the next generation with a probability $\lambda$ (default as 0.1). This creates a recovery path, ensuring that globally optimal designs can survive misclassification and re-enter the population or guide the search across disjoint feasible regions.
> 2. **Ablation Study Evidence**: The necessity of this mechanism is empirically proven in our Ablation Study (Table 4). Experiment results show that the complete CODA outperforms variant *CODA-w/o IT* in 8 out of 10 cases, indicating that the infeasible-assistant selection strategy is valuable for effective optimization.
> 3. **Final Ground-Truth Validation:** CODA does not output a single prediction. It maintains an external archive of the top 128 candidates found throughout the search, all of which undergo ground-truth evaluation at the end. This ensures that the final selection is based on actual simulation data, not surrogate predictions.
>
> [1]Yazdanbakhsh, Kumar et al. “Data-Driven Offline Optimization for Architecting Hardware Accelerators”. *2022 ICLR*
> [2]Kao, Sheng-Chun et al. "Confuciux: Autonomous Hardware Resource Assignment for DNN Accelerators Using Reinforcement Learning." *2020 MICRO*.
> [3]Kao, Sheng-Chun et al. "Gamma: Automating the HW Mapping of DNN Models on Accelerators via Genetic Algorithm." *2020 ICCAD*
> [4] Kwon, Hyoukjun, et al. "Understanding Reuse, Performance, and Hardware Cost of DNN Dataflow: A Data-Centric Approach." *2019 MICRO*
> [5]Hyoukjun, Ananda, et al. “MAERI: Enabling Flexible Dataflow Mapping over DNN Accelerators via Reconfigurable Interconnects.” *2018 ASPLOS*

---

### Official Review · Reviewer_yHeB · 2025-11-01

**Soundness:** 3
**Presentation:** 2
**Contribution:** 2
**Rating:** 4
**Confidence:** 4

**Summary:**

This work aims to design hardware accelerators efficiently while complying with a hardware constraint.

Many prior works for designing hardware accelerators have suffered from a feasibility problem that cannot guarantee the feasibility of searched solutions. It's because feasible solutions are distributed sparsely in the whole search space.
Also, it is costly to evaluate architectures because of the simulation costs.It leads to limited offline data, which is a major barrier for offline search. Besides, those huge simulation costs also hinder online search.

To resolve this problem, the paper proposes a CPC network (Constraint-Performance Cascade network) that can help both evaluate feasibility and predict hardware metrics of architectures. It consists of a feasibility classifier and a performance predictor. The feasibility classifier judges whether candidate architectures are feasible or not. The performance predictor predicts the latency of candidate architectures.

Also, to train the above framework well, the paper proposes uncertainty-weighted dual-task learning that balances feasibility classification and performance prediction. With the trained CPC network, optimized hardware accelerator architectures that are feasible and comply with the given constraint can be searched by evolutionary search.

**Strengths:**

- The paper justified itself by pointing out the limitations of prior works for designing hardware accelerators.
- The paper executed experiments well that could deliver what the paper contends.
- It is worth mentioning that verifying not only the validity of searched hardware architectures but also the performance of proposed submodules.

**Weaknesses:**

- There seems to be a lack of theoretical support in general.
- It is inevitable that a model trained with limited offline data suffers from a bias problem. There is insufficient explanation of how this problem can be overcome by the proposals.
- About uncertainty-weighted dual-task learning, the paper claims that 'the proposed mechanism alleviates data sparsity by extracting complementary information'. However, the support for this claim is not provided well.
- Typo: In page 5, multi-tak -> multi-task / te -> the

**Questions:**

- The proposed CPC network cannot help but be trained with limited offline data. When an out of distribution architecture that is infeasible comes as an input, the feasibility classifier may misclassify the architecture in some cases because it hasn't ever seen an architecture like that. Also, it is difficult for the performance predictor to predict the performance of the candidate architecture well, for the same reason. Please give the authors' opinion on whether the above problem can be solved by the proposed method.

- The review think that the explanation about uncertainty-weighted dual-task learning and equation (7) is insufficient. Can the authors provide additional explanation about them?

---

> ### Author Response · Authors · 2025-12-03
> **Response to Reviewer #yHeB (part 1/2)**
>
> We appreciate the reviewer for recognizing our clear justification of CODA's necessity, the well-executed experiments that robustly support our claims, and the importance of our thorough validation. We hope the following responses address the remaining concerns effectively.
>
> **[W1, theoretical grounding and design rationales]**
>
> We acknowledge that our work focuses on solving the empirical challenges of discrete accelerator hardware optimization. However, our framework is built upon established theoretical principles:
>
> 1. **Structural Inductive Bias:** The Cascaded-Performance Cascade (CPC) network leverages the Transformer architecture as a specific inductive bias aligned with the hardware design space. Since accelerator layers exhibit sequential dependencies regarding resource allocation and data reuse, the self-attention mechanism theoretically models these non-local interactions. Furthermore, the cascaded design imposes a structural prior that models performance as conditionally dependent on feasibility, ensuring the latent space captures constraint-relevant features first.
> 2. **Probabilistic Grounding:** The uncertainty-weighted loss in our dual-task learning (Eq. 7) is not a heuristic combination. It is **formally derived** from maximizing the Gaussian likelihood under homoscedastic task-specific uncertainty[1]. This provides a rigorous mathematical basis for dynamically balancing the distinct scales of binary feasibility classification and continuous latency regression.
> 3. **Constrained Optimization in Discrete Spaces**: Theoretical convergence proofs are generally intractable for such combinatorial, non-differentiable design spaces. Our constraint-aware evolutionary search is grounded in principles of **boundary search**. By stochastically retaining promising infeasible solutions, the algorithm exploits the boundaries of the feasible manifold to bridge disjoint feasible regions, thereby preventing premature convergence in the high-dimensional landscape.
>
> [1] Kendall, Alex et al. “Multi-task Learning Using Uncertainty to Weigh Losses for Scene Geometry and Semantics.” 2018 CVPR
>
> **[W2 & Q1, bias problem & out-of-distribution evaluation]**
>
> We acknowledge that bias from limited offline data is an inherent challenge. However, CODA mitigates this issue and achieves robust OOD performance through the following mechanisms:
>
> 1. **Regularization via Dual-Task Learning**: Unlike simple predictors, our **Collaborative Dual-Task Learning** (Section 4.1) forces the model to learn a shared representation that satisfies both feasibility classification and performance regression. This acts as a strong regularizer, preventing the model from overfitting to superficial correlations in the training set and encouraging it to capture fundamental hardware structural dependencies.
> 2. **Rank Consistency over Absolute Accuracy:**  In the optimization process, the relative ranking of candidates is more critical than absolute prediction accuracy. Even if OOD inputs introduce bias in absolute latency values, our high Spearman’s rank correlation (**0.99**, Table 7)  suggests that the model maintains the correct ordinal relationship, effectively guiding the search towards better regions despite potential bias.
> 3. **Robustness via Evolutionary Search:** We explicitly designed the search algorithm to tolerate classifier errors. By stochastically retaining a fraction ($\lambda$) of "predicted-infeasible" candidates, we prevent the search from being blocked by false negatives (misclassified feasible OOD architectures). This ensures that novel, high-potential designs are not discarded due to model bias.
> 4. **Empirical Evidence (Zero-Shot):** The effectiveness of these strategies is empirically proven by our Zero-Shot experiments (Section 5.2). CODA successfully generalizes to completely unseen dataflows (OOD scenarios) without additional training, outperforming baselines. This confirms that the learned context-aware representations are robust to distribution shifts.

---

> ### Author Response · Authors · 2025-12-03
> **Response to Reviewer #yHeB (part 2/2)**
>
> **[W3 & Q2, uncertainty-weighted learning & data sparsity]**
>
> The uncertainty-weighted mechanism is critical for extracting robust signals from sparse offline data. We provide the following clarifications regarding its theoretical basis and its role in alleviating data sparsity:
>
> - **Theoretical Basis (Eq.7)**: Equation (7) is not a heuristic design but is formally derived by maximizing the joint Gaussian likelihood of the dual objectives, following Kendall et al (2018) [1], the terms $\sigma_f$ and $\sigma_p$ represent the **homoscedastic task-specific uncertainty**. The $\log \sigma$ terms emerge naturally from the log-likelihood derivation, acting as regularization terms that prevent the uncertainties from growing indefinitely. This allows the model to dynamically scale the gradients of the feasibility and performance tasks based on their varying noise levels and units without manual tuning.
> - **Alleviating Data Sparsity**: In sparse datasets, models are prone to overfitting to the easier task (often feasibility) while neglecting the harder performance prediction (regression task), or failing to converge due to gradient conflict. Our mechanism mitigates this by:
>     1. **Enforcing Shared Representation:** By training a shared encoder for two distinct tasks, we impose an **inductive bias** that forces the model to learn structural features relevant to *both* validity and latency. This multi-view supervision increases the information density extracted from each limited sample.
>     2. **Preventing Task Dominance:** The adaptive weighting ensures that the "easier" task does not dominate the gradient updates. This effective regularization prevents the model from collapsing into a trivial solution (only predicting feasibility well), thereby maintaining the generalization capability required for the sparse design space.
>
> The effectiveness of this mechanism is validated in our ablation study (Section 5.4). As shown in Table 4, the variant without uncertainty weighting (*CODA-w/o UW*) consistently underperforms the full model (latency degrades from 33.12 to 37.95 in Case 8). This empirical gap confirms that static weighting fails to fully exploit the limited data, whereas our adaptive approach successfully extracts complementary information.

---

### Author Response · Authors · 2025-12-03
**Global Response**

Dear AC and Reviewers:

We would like to express our sincere gratitude for the time and effort the reviewers and AC have invested in reviewing our paper. First of all, we are so honored that our work has been recognized as well-justified, clear, and adaptable technical design (Reviewer #yHeB and #TbNx). We are also pleased to see the reviewers have commended CODA for its **important problem** (Reviewer #mCrF and #w4kV), **novel and well-motivated methodology** (all reviewers), **rigorous experiments** (Reviewer #yHeB, #mCrF, and #w4kV), and **better optimization performance** (all reviewers).

In this global response, we primarily summarize common suggestions shared by the reviewers and provide an overview of the additional discussion that addresses these suggestions, as follows.

**[Architectural design and underlying rationale of the CPC, Reviewer #yHeB and #w4kV ]**

Reviewer #yHeB and #w4kV collectively highlight the need for a more thorough elaboration of the methodological rationale and internal mechanisms within our framework. In response to this feedback, we have substantially revised **Section 4** of the manuscript. The revisions include a significantly expanded and clarified description of the CPC network's architecture, offering greater detail on its components and their interconnections to enhance readers’ understanding of our framework's design.

**[Qualitative analysis on prediction errors, Reviewer #yHeB, #mCrF and #w4kV ]**

Reviewers #yHeB, #mCrF, and #w4kV collectively suggested a more detailed qualitative analysis of CPC's prediction errors. In direct response to this valuable feedback, we have substantially expanded **Appendix D** to provide a comprehensive error-mode analysis. The new appendix includes: (1) an analysis of optimization trajectory and feasibility drift, (2) a fine-grained classification error analysis on the final population, and (3) a concluding discussion. This qualitative examination is designed to help readers better understand the operational strengths and inherent limitations of the CPC model within the CODA framework.

**[Other infeasibility causes/constraints, Reviewer #w4kV and #TbNx]**

Reviewer #w4kV and #TbNx suggest investigating CODA's performance under a broader range of infeasibility causes and constraints. We hence conducted additional experiments evaluating CODA in multi-constraint scenarios, where designs must simultaneously satisfy area, power, and NoC bandwidth constraints. The results, presented in **Appendix C.6, Figure 9**, demonstrate that CODA maintains highly competitive performance compared to both online and offline baselines. This finding confirms that CODA's feasibility modeling paradigm is not only conceptually natural but also remains practically effective and robust when extended to more complex, real-world constraint settings.

**[Add more ablation study, Reviewer #mCrF and #w4kV ]**

1. As Reviewer #mCrF suggested, we performed ablation studies to investigate the sensitivity of CODA's performance to the offline dataset size and the architecture of the Transformer encoders. The results, presented in **Section 5.4 (Table 3 and Table 4)**, confirm that both factors are critical for final performance. Our analysis demonstrates that the specific configuration adopted in the main experiments achieves the optimal balance, providing clear guidance for practical applications.
2. Reviewer #w4kV wants to know the influence of the search algorithm. In response, we compared our evolutionary search strategy against a random search baseline with identical feasibility filtering. The results, detailed in **Section 5.4, Table 4,** show that our evolutionary search consistently outperforms the random search baseline, achieving comparable or superior results in 9 out of 10 experimental cases. This validates the importance of a structured search strategy in efficiently navigating the complex design space.

---

### Note · Authors · 2026-03-19

I have read and agree with the venue's withdrawal policy on behalf of myself and my co-authors.

---

### Meta-Review · Area_Chair_T98M · 2026-01-19

**Summary:**

Automatic AI accelerator design is a crucial topic in supporting various AI workloads. However, the design space is very large and there are many locally infeasible regions, which were neglected by other works. Considering the irregular feasible region, given a set of AI accelerator constraint, e.g., die size, power consumption, etc., is a key. Existing training data is not often enough to train a surrogate model and therefore, it lacks the feasibility guarantee. To this end, they design a better surrogate model based on recent architectures and propose a framework of Constrained Offline Design of Accelerators (CODA).This surrogate is optimized in evaluating the feasibility and then the performance of accelerator designs. After that, they use an evolutionary search method that is able to consider constraints. They conduct experiments for various basic computer vision architectures and compare with other design methods in terms of various performance factors.

**Reviewer Concerns:**

Reviewers raised several concerns: i) the robustness of the surrogate model for out-of-distribution inputs, ii) whether end-products can be easily made from the suggested design, i.e., HDL-based output, iii) the effectiveness of the evolutionary search, iv) the availability of advanced search considering memory hierarchy and so on, and v) etc.

I think those concerns are very critical, but reviewers' rebuttal does not successfully address them. Reviewers acquire excuses for some concerns. In my opinion, however, this topic has been studied actively for the past several years and the second and fourth concerns should be resolved before the publication.

**Reviewer Scores:**

All reviewers did not explicitly leave comments, but I guess that they may not buy out the rebuttal answers made by the authors. Many rebuttal messages are for excuse purposes,

---

### Decision · Program_Chairs · 2026-01-26

Reject